# Abundant clock proteins point to missing molecular regulation in the plant circadian clock

Uriel Urquiza-García [ID] [1,3,4], Nacho Molina [ID] [1,5], Karen J Halliday [ID] [2] & Andrew J Millar [ID] [1 ✉]

## Abstract

**Understanding the biochemistry behind whole-organism traits such as flowering time is a longstanding challenge, where mathematical models are critical. Very few models of plant gene circuits use the absolute units required for comparison to biochemical data. We refactor two detailed models of the plant circadian clock from relative to absolute units. Using absolute RNA quantification, a simple model predicted abundant clock protein levels in *Arabidopsis thaliana*, up to 100,000 proteins per cell. NanoLUC reporter protein fusions validated the predicted levels of clock proteins in vivo. Recalibrating the detailed models to these protein levels estimated their DNA-binding dissociation constants ($K_d$). We estimate the same $K_d$ from multiple results in vitro, extending the method to any promoter sequence. The detailed models simulated the $K_d$ range estimated from LUX DNA-binding in vitro but departed from the data for CCA1 binding, pointing to further circadian mechanisms. Our analytical and experimental methods should transfer to understand other plant gene regulatory networks, potentially including the natural sequence variation that contributes to evolutionary adaptation.**

**Keywords** Gene Regulatory Networks; Biological Clocks; Circadian Rhythms; Mathematical Modelling; Plant Biology
**Subject Categories** Computational Biology; Plant Biology

## Introduction

Circadian clocks are intracellular regulators that control temporal gene expression patterns and hence metabolism, physiology and behaviour, from sleep/wake cycles in mammals to flowering in plants (Bass and Takahashi, 2010; Bendix et al, 2015; Millar, 2016). Clock genes are rarely essential but appropriate alleles can confer a competitive advantage (Ouyang et al, 1998; Dodd et al, 2005), have been repeatedly selected during crop domestication (Bendix et al, 2015; Muller et al, 2015) and are implicated in human disease, including cancers, metabolic and mental health (Roenneberg et al,

2022). Systems biology uses models to link these organismal traits to molecular pathways, in order to understand and potentially to engineer circadian functions (Clark et al, 2020; Chew et al, 2022). Plant biologists have a particular opportunity to connect molecular understanding to the rich tradition of crop science models (Thomas, 2007; Marshall-Colon et al, 2017; Hammer et al, 2019), alongside large-scale plant phenomics data (Tardieu et al, 2017). This study focuses on the clock gene circuit in the laboratory model plant *Arabidopsis thaliana*, as the non-transcriptional timing mechanism in this species has not been characterised (Edgar et al, 2012).

The Arabidopsis clock circuit comprises a dozen genes with tightly-interlinked feedbacks that are sufficient to generate 24-hour rhythmicity in mathematical models (see below). To simplify, dawn-expressed transcription factors *LATE ELONGATED HYPO-COTYL* (*LHY*) and *CIRCADIAN CLOCK-ASSOCIATED 1* (*CCA1*) inhibit the expression of evening genes such as *GIGANTEA* (*GI*), EARLY FLOWERING genes (*ELF3* and *ELF4*) and *LUX ARTHYHMO* (*LUX*) by directly binding to their promoter regions through the 'Evening Element' target sequence (Adams et al, 2018; Harmer and Kay, 2005; Kamioka et al, 2016; Nagel et al, 2015). *LHY* and *CCA1* expression is ended by the binding of repressors from the *PSEUDO-RESPONSE REGULATOR* (*PRR*) gene family, which are expressed in the day, in sequence *PRR9*, *PRR7*, *PRR5* and *TOC1* (*TIMING OF CAB2 EXPRESSION 1*) (Nakamichi et al, 2010; Huang et al, 2012; Gendron et al, 2012). Falling LHY and CCA1 protein levels allow the expression of ELF3, ELF4 and LUX proteins that form an "Evening Complex" in the early subjective night (Nusinow et al, 2011). *Via* the LUX subunit (also known as PHYTOCLOCK1), the complex binds to and represses the expression of *TOC1* and several evening genes (Helfer et al, 2011; Silva et al, 2016), while the PRR proteins degrade, allowing *LHY* and *CCA1* expression in the late night to start the cycle anew. The pace of this repressor-based circuit is modified by transcriptional activators (Perez-Garcia et al, 2015; Shalit-Kaneh et al, 2018; Urquiza-García and Millar, 2021) and by post-translational regulation, including from light input pathways, which entrain the clock to ensure that rhythmic activities occur at appropriate phases relative to the external, day/night cycle (Millar, 2016).

A series of mathematical models has represented the biochemistry of the clock gene circuit with increasing detail in differential equations (Pokhilko et al, 2012, 2013; Fogelmark and Troein, 2014;

[1]Centre for Engineering Biology and School of Biological Sciences, C. H. Waddington Building, University of Edinburgh, King's Buildings, Edinburgh EH9 3BF, UK. [2]School of Biological Sciences, Daniel Rutherford Building, University of Edinburgh, King's Buildings, Edinburgh EH9 3BF, UK. [3]Present address: Institute of Synthetic Biology, University of Düsseldorf, Düsseldorf, Germany. [4]Present address: CEPLAS-Cluster of Excellence on Plant Sciences, Düsseldorf, Germany. [5]Present address: Institut de Génétique et de Biologie Moléculaire et Cellulaire (IGBMC) CNRS UMR 7104, INSERM U964, Université de Strasbourg, 1 Rue Laurent Fries, 67404 Illkirch, France. ✉E-mail: andrew.millar@ed.ac.uk

Urquiza-García and Millar, 2021), while other models used simpler versions (Dalchau et al, 2011; De Caluwé et al, 2016; Foo et al, 2020; Greenwood et al, 2022). The models have previously predicted new molecular clock components and interactions (reviewed in Bujdoso and Davis, 2013), and explained some operating principles of the clock mechanism (Akman et al, 2008; Edwards et al, 2010; Gould et al, 2013; Rand, 2008). Modelling the circadian control of downstream pathways (Seaton et al, 2015) has allowed us to bridge the genotype-phenotype gap, linking molecular regulation to whole-organism traits (Chew et al, 2022). However, the detailed models have two general limitations. First, they used real time units but arbitrary mass units, so the values of many biochemical-kinetic parameters could not be validated against biochemical data, such as synthesis rates or binding constants. One exception recently introduced absolute RNA levels (Urquiza-García and Millar, 2021). Second, the genes in these models are functional units with no internal structure, so they cannot directly represent genetic variation at the level of genome sequence. Here, we refactor the Arabidopsis clock model to represent plant clock protein levels in absolute units of protein copies per cell, introduce tractable methods that facilitate validation against biochemical and in vivo data, and extend one method to include genome sequence directly.

Clock protein numbers have been both measured and modelled in the fungus *Neurospora crassa* (Merrow et al, 1997; Smolen et al, 2003), in mammalian cells (Gabriel et al, 2021; Koch et al, 2022; Kramer et al, 2020; Narumi et al, 2016; Smyllie et al, 2016) and in the cyanobacterium *Synechococcus elongatus* (Chew et al, 2018; Kitayama et al, 2003). The absolute numbers of proteins directly constrain their possible biochemical activities (Kim and Forger, 2012) and dynamic behaviour (Leise et al, 2012; Chew et al, 2018; Gould et al, 2018; Jeong et al, 2022a), refining our understanding of clock mechanisms. Consider, for example, a clock transcription factor that rhythmically binds to the promoters of target genes. A protein with a high DNA-binding affinity (low dissociation constant, $K_d$) cannot be such a rhythmic regulator, if it is active at concentrations well above the $K_d$ at all circadian phases, binding to its target sites throughout the day and night. Hence the levels of the clock proteins constrain their possible dissociation constants and vice versa, if the protein is to function as expected, in this case as expected in a mathematical model. The plant clock models include this mutual constraint but their $K_d$ values cannot be compared to measured binding data, because the models' $K_d$ values have arbitrary units.

## Clock protein levels as a Fermi problem

We approach this question as a "Fermi problem" (Phillips and Milo, 2009), combining the available data from diverse sources to give coarse predictions for the numbers of plant clock protein molecules per cell in *Arabidopsis thaliana* (Fig. 1). We rescale the detailed clock models for these protein levels, which also returns simulated $K_d$ values for their DNA binding in absolute units. The model predictions are compared with data-driven estimates of these dissociation constants, using an approach that allows promoter sequence data to inform the clock models. Lastly, we use reporter fusion proteins to measure the clock protein levels, largely validating our initial predictions, in plant extracts and by in vivo imaging.

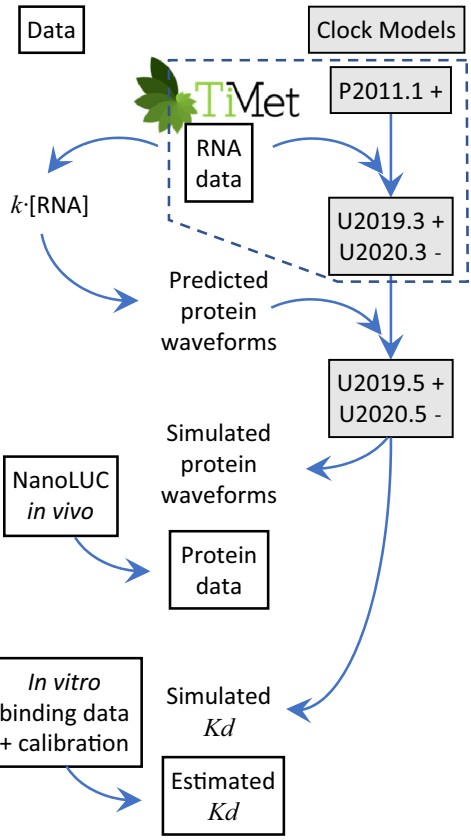

**Figure 1. Model calibration and testing.**

Models of the Arabidopsis circadian clock (shaded boxes), with PRR activation (+) or inhibition (−), were calibrated to match experimental data (open boxes). Absolute RNA levels from the TiMet project were previously used to calibrate the U2019 and U2020 models (dashed outline; Urquiza-García and Millar, 2021). Here, a simple protein model ($k\cdot[\text{RNA}]$) uses the same RNA data to estimate absolute protein levels and recalibrate new versions of each model. The models first simulate detailed protein waveforms that are tested against absolute protein data from NanoLUC reporter genes in transgenic plants. Second, the models predict transcription factor dissociation constants for DNA-binding ($K_d$) in vivo. These are tested against the constants estimated in vitro from surface plasmon resonance data, protein-binding microarrays, genome sequences transformed by a binding energy matrix, ChIP-seq and nuclear volume data.

## Results

### Predicting clock protein levels from mRNA data in absolute units

We extended an earlier method that was applied to predict metabolic enzyme levels in moles per gramme fresh weight (Piques et al, 2009), in order to predict the levels of clock proteins in units of molecules per cell. Timeseries of mRNA transcript levels for multiple clock proteins were previously measured in molecules per cell (Fig. 2A), using calibrated qRT-PCR assays (Flis et al, 2015). If the relevant translation and degradation rates are known, protein levels can be estimated from these RNA data using a simple, data-driven model,

$$\frac{dP}{dt} = sM(t_i) - kP \qquad (1)$$

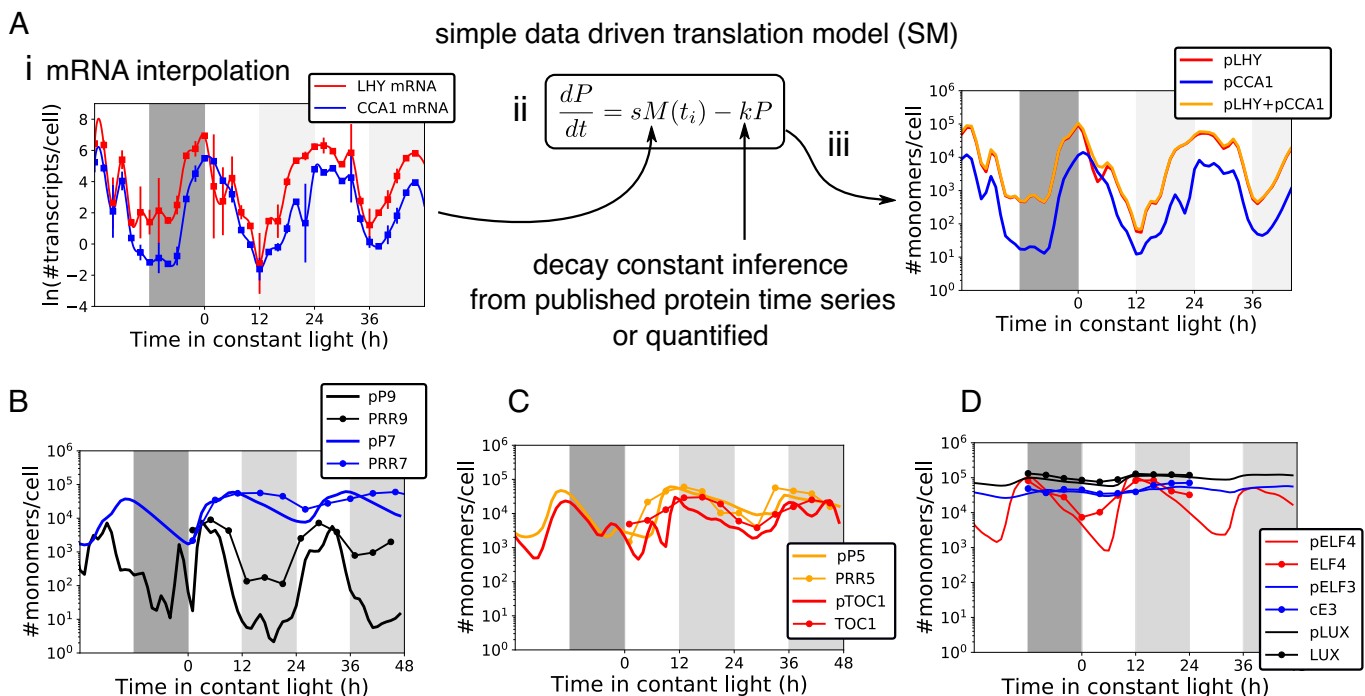

**Figure 2. Predicting clock protein levels from RNA data.**

(A) i. mRNA timeseries data from the TiMet project (Flis et al, 2015) was log-transformed and interpolated with a cubic spline (*LHY* mRNA, red; *CCA1* mRNA, blue). ii Interpolated RNA data *M(t)* was the input to the simple model. The translation rate *s* was calculated and protein decay constants *k* were derived from literature, as described in Results. Protein timeseries predicted from the simple model (solid lines without markers, protein names with prefix p) in absolute units are plotted with protein timeseries from the literature (with markers), for (A.iii) LHY and CCA1, (B) PRR9 and PRR7, (C) PRR5 and TOC1 and (D) ELF3, ELF4 and LUX. Literature data in arbitrary units were rescaled to the maximum value of the model prediction within each timeseries, to compare protein waveforms. (Aiii–D) have the same logarithmic scale.

Where *P* stands for protein amount, *s* for translation rate, $M(t_i)$ mRNA transcript level in absolute units at time $t = i$ and *k* is the decay constant of the protein of interest (Fig. 2B). We follow past work (Piques et al, 2009) in estimating the translation rate *s* for each protein, from a ribosome elongation rate measured in mammalian cells, corrected for the lower growth temperature of Arabidopsis plants, to give 3 codons/s, using the measured ribosome density of 6.6 ribosomes per kb mRNA transcript from *E. coli*, and lengths of open reading frames from the TAIR10 genome annotation (see Appendix). Effective degradation rates *k* were taken from measured protein decay curves for LHY and CCA1 proteins (Hansen et al, 2017; Song and Carré, 2005), and for PRR7, PRR9, PRR5 and TOC1 proteins (Farre and Kay, 2007; Ito et al, 2007; Kiba et al, 2007; Mas et al, 2003). Decay timeseries data for the Evening Complex proteins has not been reported, so the effective rates were estimated by fitting (see Appendix) to profiles derived from published protein gel blot images (Nusinow et al, 2011). We also digitised full timeseries profiles under LD and/or LL for the above-mentioned proteins from the literature, using the same approaches. Light strongly destabilises some clock proteins, so separate, light and dark estimates were used for some degradation rates in the data-driven model (Table EV1), whereas the smaller effect of light on translation rate was ignored (Kim et al, 2003; Piques et al, 2009).

The predicted protein timeseries closely followed the mRNA profiles (compare Figures 2Ai,Aiii). Table 1 shows that the predicted levels for the clock proteins of interest peaked at around 100,000 protein copies per cell and fell to trough levels in the range of 1000–5000 protein copies per cell (plotted in Fig. 2B–D). PRR9 had a lower predicted level due to its low RNA level (Fig. 2B). The predicted amplitude of protein regulation changed in constant light conditions (LL) compared to light:dark cycles, for example from a 32-fold amplitude of predicted PRR7 protein rhythmicity in LD to 5-fold amplitude under LL (Table 1). This reflected the lower amplitude of mRNA rhythms observed in general under LL (Flis et al, 2015) (Fig. 2B). The timing of the simple model's predicted waveforms closely matched the peaks of the protein gel blot profiles (Figure 2Aiii–D), though PRR7 had a broader peak during the subjective night in the data (Fig. 2B). Amplitude information is hard to derive by digitising gel blot images from the literature.

## Detailed models with absolute protein copy numbers

Our detailed models of the clock gene circuit (see Introduction) are not driven by rhythmic data input like the simple model, but rather use ordinary differential equations to recapitulate the dynamics of each RNA and protein component in the clock circuit, along with their interconnected feedback loops and their regulation by light signals. The models autonomously generate rhythmic patterns of RNA and protein expression that match the rhythmic data. The gene circuits in models U2019 and U2020 differ only in the regulation of daytime processes, involving *LHY/CCA1* and the *PRR* genes (Urquiza-García and Millar, 2021). The circuit of U2019 is

**Table 1. Protein levels predicted by the simple model, simulated by the full models and observed in extracts or in vivo.**

| Protein | Conditions: Model or data | LD | | | | LL | | | | Plot |
|---|---|---|---|---|---|---|---|---|---|---|
| | | Peak | Trough | Fold-change | ZT LL | Peak | Trough | Fold-change | LL/LD ratio | |
| CCA1 + LHY | Pred. | 89,506 | 460 | 194.6 | 12-34 | 58,786 | 449 | 130.9 | 0.67 | Fig. 2Aiii |
| CCA1/LHY | Sim. U2019.5 | 103,803 | 10,041 | 10.3 | 12-34 | 53,061 | 7825 | 6.8 | 0.66 | Fig. 3B LHY + CCA1 panel |
| CCA1/LHY | Sim. U2020.5 | 114,645 | 8410 | 13.6 | 12-34 | 37,379 | 6326 | 5.9 | 0.43 | Appendix Fig. S1A |
| LHY-NL | Obs. in vitro | 121,118 | 1431 | 84.6 | 12-34 | 188,712 | 3484 | 54.2 | 0.64 | Fig. 6A |
| CCA1-NL | Obs. in vitro | 182,067 | 4682 | 38.9 | 12-34 | 93,877 | 2816 | 33.3 | 0.86 | Fig. 6B |
| PRR7 | Pred. | 37,341 | 1613 | 23.2 | 24-48 | 59,200 | 7605 | 7.8 | 0.34 | Fig. 2B |
| PRR7 | Sim. U2019.5 | 42,004 | 3124 | 13.4 | 24-48 | 28,623 | 5558 | 5.1 | 0.38 | Fig. 3B PRR7 panel |
| PRR7 | Sim. U2020.5 | 54,589 | 152 | 359.1 | 24-48 | 13,885 | 2829 | 4.9 | 0.01 | Appendix Fig. S1B |
| PRR7-NL | Obs. in vitro | 81,208 | 4761 | 17.1 | 24-48 | 55,881 | 3875 | 14.4 | 0.85 | Fig. 6B |
| TOC1 | Pred. | 23,313 | 493 | 47.3 | 12-34 | 29,981 | 1877 | 16.0 | 0.34 | Fig. 2C |
| TOC1 | Sim. U2019.5 | 33,978 | 3872 | 8.8 | 12-34 | 48,752 | 3221 | 15.1 | 1.72 | Fig. 3B TOC1 panel |
| TOC1 | Sim. U2020.5 | 14,759 | 2639 | 5.6 | 12-34 | 14,443 | 2342 | 6.2 | 1.10 | Appendix Fig. S1C |
| TOC1-NL | Obs. in vitro | 44,482 | 3523 | 12.6 | 12-34 | 67,995 | 3523 | 19.3 | 1.53 | Fig. 6C |
| ELF3 | Pred. | 44,868 | 26,280 | 1.7 | 24-48 | 57,426 | 29,879 | 1.9 | 1.13 | Fig. 2D |
| ELF3 | Sim. U2019.5 | 19,813 | 2783 | 7.1 | 24-48 | 28,035 | 7212 | 3.9 | 0.55 | Fig. 3B ELF3 panel |
| ELF3 | Sim. U2020.5 | 11,319 | 516 | 21.9 | 24-48 | 14,705 | 2827 | 5.2 | 0.24 | Appendix Fig. S1D |
| ELF3-NL | Obs. in vivo | 41,798 | 24,897 | 1.7 | 24-48 | 28,877 | 17,892 | 1.6 | 0.96 | Fig. 6F |
| LUX | Pred. | 99,830 | 59,001 | 1.7 | 24-48 | 122,351 | 57,535 | 2.13 | 1.26 | Fig. 2D |
| LUX | Sim. U2019.5 | 167,943 | 15,502 | 10.8 | 24-48 | 165,069 | 12,957 | 12.74 | 1.18 | Fig. 3B LUX panel |
| LUX | Sim. U2020.5 | 145,379 | 6949 | 20.9 | 24-48 | 138,629 | 10,261 | 13.51 | 0.65 | Appendix Fig. S1G |
| LUX-NL* | Obs. in vivo* | 4455 | 1577 | 2.8 | 24-48 | 6355 | 1807 | 3.52 | 1.24 | Fig. EV1F |
| LUX-NL* | Obs. in vitro* | 596 | 379 | 1.6 | 24-48 | 882 | 362 | 2.4 | 1.55 | Fig. EV2C |

Peak and trough levels are given as the protein copy number per cell, along with relative amplitude (fold change). Levels under 12L:12D cycles (LD) are all from ZT0-24h of the relevant timeseries. Levels in constant light conditions (LL) are from the cycle indicated (ZT LL). LL/LD ratio is the relative amplitude in LL compared to LD. Simulated levels (Sim.) combine all pools of each protein. Levels were observed (Obs.) using NanoLUC fusion reporters (NL) either in plant extracts (in vitro assay) or in vivo. Values for LUX are included for completeness; *measurements for LUX-NL are preliminary due to partial complementation by the fusion protein.

closer to its antecedent model P2011 (Pokhilko et al, 2011), using gene activation, whereas U2020 uses repression (for circuit diagrams, see Supplementary Fig. 3 of (Urquiza-García and Millar, 2021). Repression is better supported by molecular data but U2020 simulations fit the data no better than, or slightly worse than, the activation-based model U2019 (consistent with Fogelmark and Troein, 2014), so we use both circuits here.

Versions U2019.3 and U2020.3 rescaled only the clock's RNA variables into absolute units (Urquiza-García and Millar, 2021), using the same RNA timeseries data from the TiMet project as the simple model, above. We could now rescale these mechanistic models from protein levels in arbitrary units to the absolute units predicted in Table 1 (equations in Supplementary Files 1, 2; for SBML model files, please see Data availability). The scale of all protein variables in the models is uniquely defined by a subset of proteins, the combined LHY/CCA1 protein, PRR7 and LUX. The PRR proteins and TOC1 function additively as transcriptional repressors in the model, so rescaling any one of them sets the scale of the other three. The stoichiometry of the Evening Complex also couples the scale of its components but of these, only LUX binds to DNA and provides DNA-binding affinity data to compare with the model prediction. The three protein scaling factors were estimated

numerically by fitting the model simulations to the protein timeseries predicted from the simple model (Fig. 3A; LHY/CCA1 was fitted to total predicted LHY + CCA1; see Methods).

The resulting models U2019.4 and U2020.4 retain the same dynamic behaviour as the parent models but simulate the clock proteins in units of molecules per cell. The version number U2019.4 indicates a model with the U2019 regulatory circuit but different parameter values from U2019.3, for example. Comparing the simple, data-driven model and the rescaled, full model, we found similar protein levels and dynamics in most cases, with expected departures where the regulation differed between the simple and full models (Fig. 3B; Appendix Fig. S1). The full models' predicted amplitudes for the PRR9 or LHY/CCA1 protein rhythms did not fully match the very high amplitudes of RNA regulation that drive the simple model, for example. More strikingly, the best-fit rescaling in both U2019.4 and U2020.4 left the mean level of ELF3 protein an order of magnitude lower than predicted from the RNA data (Fig. 3B; see Discussion). The simple model's predicted protein levels alone were potentially informative but the detailed models link the protein levels to more detailed timeseries simulations, and also to the proteins' functions in DNA binding.

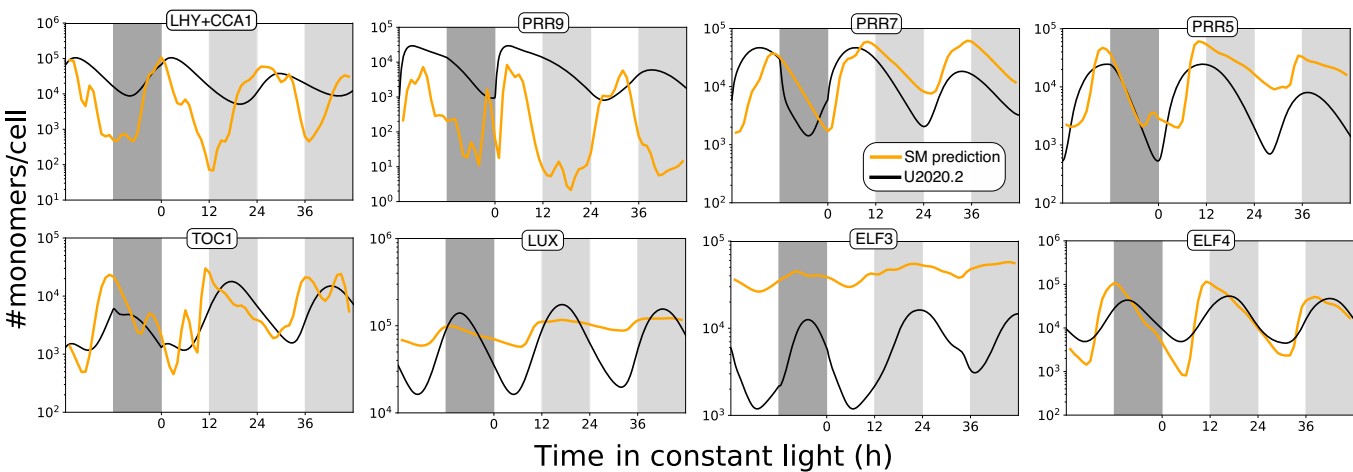

**Figure 3.  Rescaling protein levels in the mechanistic models.**

Scaling factors were introduced into the protein equations to form models U2019.4 and U2020.4, in order to match the protein levels predicted by the simple model rather than the arbitrary units of models U2019.3 and U2020.3. In (**A**), variable *cL* (LHY/CCA protein) in arbitrary units is replaced by $cL_s$ in molecules per cell, by introducing scaling parameter *s*. This example shows the equation for *TOC1* mRNA $cT_m$, where *cEC* is the Evening Complex repressor copy number; *g, m* and *n* parameters are listed in Tables EV2 and EV3. In U2019.5 and U2020.5, all parameters are rescaled, so $g_s$ takes the value $s{\cdot}g_s$. (**B**) Fitting the simulated proteins to the protein waveforms predicted from the simple model (orange lines) estimated the best-fit values of the scaling factors. The resulting models U2019.4 (**B**) and U2020.4 (Appendix Fig. S1) simulate protein dynamics in molecules per cell (black lines). Simulated ELF3 protein remained below the predicted levels. The models were entrained to ten 12L:12D cycles prior to the interval plotted. Plots are scaled differently for each protein; Fig. 2 shows all predicted proteins on the same scale.

## Comparing model predictions to measured DNA-binding affinities

Models U2019.5 and U2020.5 propagated the rescaling from the scaling factors alone to all the model parameters (Tables EV2 and EV3), including the clock proteins' dissociation constants for DNA-binding ($K_d$). The rescaled models predicted relatively high $K_d$ values (low binding affinities) for LHY/CCA1 in the 100 nM range (Table 2, Table EV4). These are within the 0.167–700 nM range of measured dissociation constants that were previously collated from the literature for plant DNA-binding proteins (Millar et al, 2019). $K_d$ values that fall within the range of protein levels simulated in the models indicate a substantial change in binding over the circadian cycle, for example in the binding of LHY/CCA1 to the *PRR9* promoter (Fig. 4). However, these simulated $K_d$ values are two orders of magnitude higher than the lowest $K_d$ measured by Surface Plasmon Resonance, 1.44 +/− 0.2 nM, for bacterially-expressed LHY/CCA1 heterodimers binding to a consensus

Evening Element sequence (O'Neill et al, 2011). 1.44 nM corresponds to ~100 proteins per nucleus. One further analysis recalibrated the simulated $K_d$ to account for ~1000 genome-wide CCA1 binding sites that are absent from the model (please see Discussion).

The detailed models predict low nM values for LUX binding to its promoter targets as part of the Evening Complex (EC; Table 2). These $K_d$'s lie within the range of simulated EC levels (Fig. 4), consistent with phase-specific binding of the EC in the model. The simulated $K_d$ are below or overlapping the 6.5 to 43 nM range of $K_d$'s measured for the LUX DNA-binding domain on consensus target sequences (Silva et al, 2016, 2020). Hence the models' simulated $K_d$'s were much closer to the values measured in vitro for LUX, where only the LUX protein in the EC can bind its targets, than the equivalent data for LHY and CCA1. However, the simulated EC copy number ranges from tens to thousands per cell (Fig. 4), at least an order of magnitude less than the total LUX. No such $K_d$ data were available for PRR proteins.

**Table 2. Clock protein dissociation constants for DNA-binding, from models or data.**

| Model regulator | Model target | parameter | Sim. $K_d$ (copy number/cell) | Sim. $K_d$ (nM) | $K_d$/ChIP peaks (nM) | Clock protein | Target gene | EMA $K_d$ (nM) |
|---|---|---|---|---|---|---|---|---|
| cEC | cE4m | g2 | 596 | 8.6 | 0.18 | LUX | ELF4 | 1.68 |
| cEC | cGm | g14 | 206 | 3.0 | 0.06 | LUX | GI | 2.49 |
| cEC | cP9m | g8 | 408 | 5.9 | 0.12 | LUX | PRR9 | 2.54 |
| cEC | cTm | g4 | 451 | 6.5 | 0.14 | LUX | TOC1 | 3.17 |
| cL | cE3m | g16 | 28,141 | 408 | 0.26 | CCA1 | ELF3 | N/A |
| cL | cE4m | g6 | 24,508 | 355 | 0.23 | CCA1 | ELF4 | 0.25 |
| cL | cGm | g15 | 42,247 | 612 | 0.40 | CCA1 | GI | 0.58 |
| cL | cLUXm | g6 | 24,508 | 355 | 0.23 | CCA1 | LUX | 0.34 |
| cL | cP9m | g9 | 33,342 | 483 | 0.31 | CCA1 | PRR9 | 0.32 |
| cL | cTm | g5 | 20,979 | 304 | 0.20 | CCA1 | TOC1 | 0.41 |
| cL+cLmod | cP7m | g10 | 67,505 | 978 | 0.63 | CCA1 | PRR7 | 0.32 |
| cLmod | cP5m | g12 | 16,905 | 245 | 0.16 | RVE8 etc. | PRR5 | N/A |

Examples of simulated $K_d$'s in model U2019.5 are listed by the protein variable, such as cEC for the Evening Complex, the regulated variable, such as cE4m for ELF4 mRNA, and the name of the $K_d$ parameter, such as g2. $K_d$/ChIP peaks, a possible calibration of the simulated $K_d$ by the number of genomic regions bound in ChIP-seq data, see Discussion. $K_d$'s derived using EMA from experimental data are listed by protein and target gene. N/A, ChIP-seq data indicate no binding to ELF3; the activator cLmod presumably corresponds to RVE8 and other proteins, for which no binding data are available.

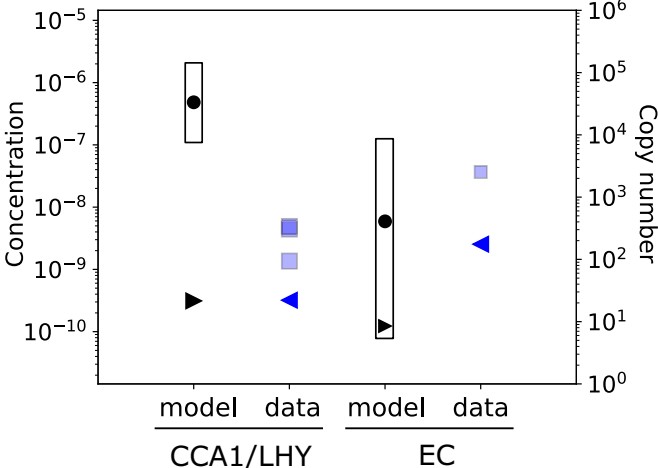

**Figure 4. Protein function in DNA binding, in simulations and data.**

The plots show the dynamic range (open bars) of the combined CCA1 and LHY protein levels (Table 1) and simulated CCA1/LHY protein (cL; left), or estimated LUX levels (Table 1) with simulated Evening Complex (EC, cEC; right) levels from model U2019.5 under 12L:12D cycles, in units of molecules per cell (right axis) or nuclear concentration (left axis). The simulated $K_d$'s for binding of each protein to PRR9 (parameters g9 and g8, respectively) fall within each protein's range of simulated levels (black dot). One possible calibration for the number of ChIP-seq targets in the genome yields the black triangles (pointing right; see Discussion). Blue squares represent the $K_d$ measured in vitro for CCA1, LHY, or mixed CCA1 and LHY (forming heterodimers; note the markers overlap), or for the LUX DNA-binding domain in the EC, each tested on its minimal, consensus binding sequence from the PRR9 promoter. Blue triangles (pointing left) show the estimated $K_d$'s for CCA1 or LUX binding to the whole PRR9 upstream region. The dynamic ranges and $K_d$'s for other genes are shown in Tables 1, 2 and Table EV4.

Among the differences between the rescaled models and the measured $K_d$'s (see also Discussion; Dataset EV1), the model $K_d$'s necessarily reflect the binding of the clock protein over the entire target gene, because the model's genes have no internal structure. In contrast, the $K_d$'s measured in vitro are for individual, high-affinity, consensus sequences. A target gene might have several exact copies of the consensus sequence or none. We therefore developed an approach to extrapolate the dissociation constants for arbitrary promoter sequences, using large-scale data that is available for *Arabidopsis thaliana*.

## Extending the approach to promoter sequences

O'Neill et al (2011) measured the $K_d$ of LHY and/or CCA1 binding in vitro to only three target sequences, two high-affinity sites and a "non-binding", mutated sequence (Harmer and Kay, 2005). Similar data for LUX were obtained by Electromobility Shift Assay (EMSA) (Silva et al, 2016, 2020). Fortunately, protein binding microarray (PBM) assays (Fig. 5A; Berger and Bulyk, 2009) have been applied to test the binding of CCA1 and LUX to all possible 8-mer sequences (Franco-Zorrilla et al, 2014; Helfer et al, 2011), and we applied a more advanced analysis than previously (see below; Fig. 5B,C). Unfortunately, their results in Enrichment score (E-score) units were not directly comparable to biophysical $K_d$ measurements. Comparing the CCA1 E-scores for the three target sequences tested by O'Neill et al, their $K_d$'s suggested a simple, linear relationship between the microarray results and the biophysical assays (Fig. 5D). Applying this linear regression to the microarray dataset would assign tentative dissociation constants for CCA1 binding to any 8-mer sequence. We performed a similar analysis for LUX, where the dissociation constants for several consensus binding sequences were obtained by Electromobility Shift Assay (EMSA) (Silva et al, 2016, 2020). The linear relationship observed there also (Fig. 5E) supported our approach of using PBM data as a proxy for $K_d$ (see Appendix).

A technical concern arose, because the "non-binding", mutated version of the high-affinity CCA1 target sequence had a measured E-score around 0.25, indicating greater CCA1 binding than to many other sequences. Experimental error in the microarray assay is expected to further confound signals, where weak binding and non-binding sequences overlap in this part of the E-score distribution (Fig. 5B). To avoid introducing an arbitrary threshold

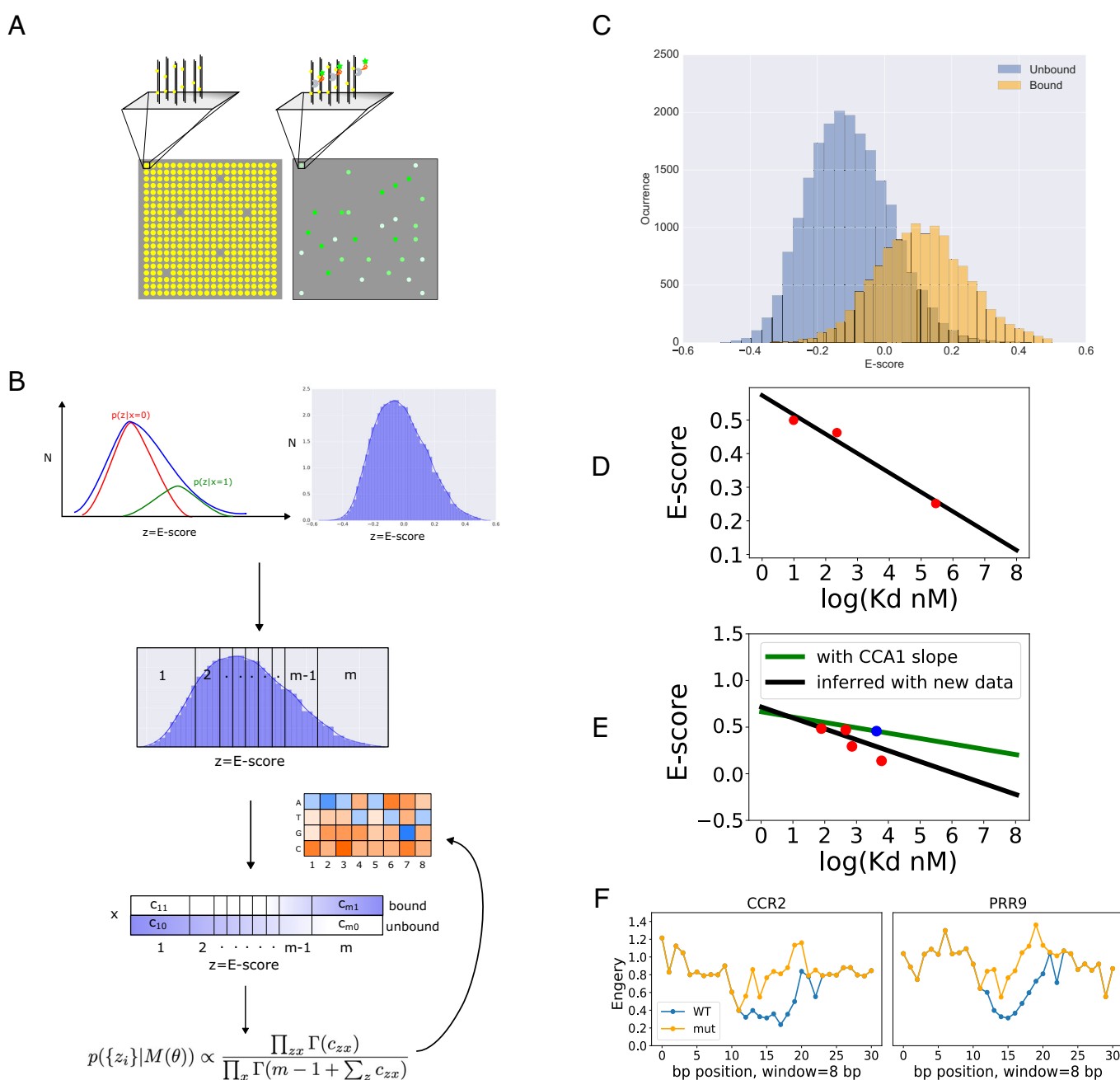

**Figure 5. Interpolating from measured $K_d$'s to any binding sequence.**

(A–C) outline the approach to estimate $K_d$. (A) Schematic of a Protein Binding Microarray (PBM, left) and the fluorescence data from binding tagged proteins (green, right), which are quantified (B) as an E-score for each 8-mer DNA binding sequence (blue line). The PGM signal distribution comprises overlapping distributions of E-scores from unbound (red line) and bound (green line) sequences. These can be separated using the Error Model Averaging algorithm (EMA; Kinney et al, 2007), which considers the E-score distribution in bins. EMA infers a best-fit matrix of binding energies, for each nucleotide at each position of the 8-mer DNA binding sequence (rectangle) and classifies bound (C, yellow) and unbound sequences. The published PBM data included the sequences used to measure the binding $K_d$ (in nM units) in vitro for CCA1 (D) and LUX (E). Only these sequences have binding data (red points) both in E-score and nM units, defining linear relationships (solid lines) that can recalibrate the E-score of any other sequence to a $K_d$ in nM. (E) also plots a potential alternative calibration (green line), applying the slope of the CCA1 calibration line to a single $K_d$ for LUX (blue point; Silva et al, 2016). (F) shows the predicted energy of CCA1 binding to an 8-basepair sequence window, in the promoters of *CCR2* (left) or *PRR9* (right) around canonical binding sites (blue line) or mutated versions (orange line).

between binding and non-binding sequences, the biophysical Error Model Averaged (EMA) approach (Kinney et al, 2007; Barnes et al, 2019) was applied (Fig. 5C) to deconvolve the distributions of bound and unbound sequences in the E-scores (Fig. 5D), using a

binding energy matrix inferred for the 8 base-positions x 4 bases in each possible target sequence on the microarray (see Appendix).

Finally, the sequences predicted to bind CCA1 (or LUX) were identified upstream of each gene in the model. The total $K_d$ was

calculated by adding the contribution of their constituent 8-mers, such that this $K_d$ estimate is directly informed by the promoter sequences (Fig. 5F). The approach might be refined to account for mutual interference among DNA-binding proteins, nucleosomes (Sullivan et al, 2014; Zhang et al, 2015) and other factors in future (Dataset EV1). The result is a $K_d$ for each target gene derived from direct DNA-binding data for CCA1 and LUX, which can be compared to the $K_d$ values simulated independently by the models in absolute units (Table 2, Table EV4).

These total $K_d$ values were typically an order of magnitude lower than the values reported for the consensus sequences alone (Fig. 4, Table 2). For LUX, the resulting estimated $K_d$'s of 1.3–3.2 nM (median 2.5 nM) were all within the models' 0.3–8.6 nM range of simulated $K_d$'s (median 5.3 nM), suggesting remarkable agreement between data and model. For CCA1, these lower $K_d$ values for the target clock genes were still further from the models' simulated $K_d$ values (Table 2, Table EV4). The $K_d$ values in U2019.5 and U2020.5 depend on the protein levels predicted by the simple model. If the predicted LHY and CCA1 levels were too high, the full models would simulate incorrectly-high $K_d$ values, so we tested those predictions by measuring clock protein levels in vivo.

## Absolute quantification of clock proteins using NanoLUC reporter fusions

Few antibodies are available for the plant clock proteins, and their low abundance has made them challenging to quantify by mass spectrometry, though each of these approaches holds promise for the future. Our implementation of the small, ostracod-derived, Nano-luciferase (NanoLUC) reporter protein in transgenic plants (Urquiza-García and Millar, 2019) suggested an alternative strategy, using calibrated luciferase assays to test the levels of reporter fusions to the clock proteins. To retain alternative splicing and other post-transcriptional regulation (James et al, 2018), we inserted a dual-epitope-tagged, NanoLUC-3xFLAG-10xHis reporter (NL3F10H) into genomic clones of the clock genes *LHY, CCA1, PRR7, TOC1, LUX* and *ELF3*, encoding translational fusions between each clock protein and a C-terminal NanoLUC (Appendix Fig. S2; Tables EV5, EV6). After testing reporter activity in transient, protoplast transfections (Appendix Fig. S3), stable transgenic lines were generated in the background of the cognate mutant (or double mutant) for each clock gene. Lines were selected by quantitative complementation of the clock-mutant phenotype (Fig. EV1), implying that the fusion protein's expression level was functionally similar to the wild-type protein. Complementation was tested using the circadian period of a transcriptional, firefly luciferase (LUC+) reporter transgene and/or a physiological, hypocotyl elongation assay (see Appendix; Table EV7).

The selected, transgenic lines were grown under light:dark cycles (LD) for 21 days and sampled every 2 h, followed by a transfer to constant light (LL), in conditions similar to the TiMet RNA timeseries study. Known amounts of purified, recombinant NanoLUC were used to calibrate in vitro assays of whole-cell extracts from a measured fresh weight of transgenic plant material, yielding rhythmic timeseries data for LHY, CCA1, PRR7 and TOC1, in units of molecules per cell (Fig. 6A–C). The measured levels of the fusion proteins were very similar to those predicted by the simple model, with peak protein expression of LHY, CCA1 and PRR7 around 100,000 molecules per cell under LD conditions and trough levels in the low to mid 1000's (Table 1). TOC1 peaked only slightly lower, at 44,000 molecules per

cell. These direct experimental measurements validated the simple model's predicted protein levels.

The LUX fusion line tested in extracts showed much lower expression levels (mid-100s of molecules per cell; Fig. EV2C), and contrasted with the strong expression of a closely-related BOA protein reporter (Urquiza-García and Millar, 2019). Further testing showed that many LUX fusion lines restored the rhythmic expression of our screening marker in the *lux-2* mutant host, but failed to complement its long-hypocotyl phenotype (Fig. EV1E). A second LUX protein reporter line that complemented both mutant phenotypes was therefore selected and tested by in vivo luminometry, along with a reporter for ELF3, as described below.

## High-resolution timeseries data compares measured and modelled plant clock proteins

The timeseries profiles of the fusion transgenes were very similar to earlier gel blot data and slightly lagged the RNA timeseries, confirming that the reporters were appropriately regulated, with peak levels of LHY at ZT2, CCA1 at ZT2-4, PRR7 at ZT10 and TOC1 at ZT16 under LD conditions (Fig. 6A–C). LHY and CCA1 protein levels remained low during an extended trough of 6–10 h in the late day to early night, while PRR7 and TOC1 levels peaked. Low signal variation among the biological triplicate samples suggested that the technical data quality from the in vitro NanoLUC assays was comparable to the TiMet qRT-PCR timeseries, allowing detailed interpretation of the protein timeseries.

Comparing the protein data to the predicted protein levels in the full model (Fig. 6), the LHY and CCA1 protein data are similar in timing to the simulated CCA1/LHY and overlap in levels, though the data has a longer, lower protein trough in LD. The measured PRR7 waveform is shifted about 2 h later than the model under LD, but with similar protein levels to the models under LL. PRR7 showed less protein degradation in the dark than predicted in the U2020.5 model; the protein data were closer to the U2019.5 profile. Phase plane diagrams compared the modelled (Fig. 7A) and measured TiMet timeseries for *LHY* and *CCA1* mRNA to the TOC1 protein timeseries (Fig. 7B). Most strikingly, their mRNA levels rise only after TOC1 levels have fallen substantially in the dark night of an LD cycle. Under LL, however, the *LHY* and *CCA1* mRNA levels rise from trough to peak while TOC1 protein is at peak levels, whereas the model shows simple, reciprocal regulation at this phase (Fig. 7A), indicating that the bulk TOC1 protein measured by the reporter fusion does not match the TOC1 function represented in the full model (see Discussion).

## Protein timeseries data from in vivo imaging

Along with the four clock reporter fusions lines described above (Figs. 6D,E and EV3), and LUX (Fig. EV1F), ELF3 reporter lines were created as outlined above and tested in vivo in seedlings. The in vivo luminescence levels from the LHY, CCA1, PRR7 and TOC1 reporters peaked within a 5-fold range, ranked LHY ~ CCA1 > PRR7 > TOC1, consistent with the peak signals in extracts (Table 1). The bioluminescence signal in vivo peaked in phase with the timeseries in extracts for LHY and CCA1, or 2 h later for PRR7 at ZT12 rather than ZT10 in vitro. The in vivo assay showed rapid falls of 10–30% in PRR7 and TOC1 fusion signals immediately after dusk (Figs. 6E and EV3). These were not evident

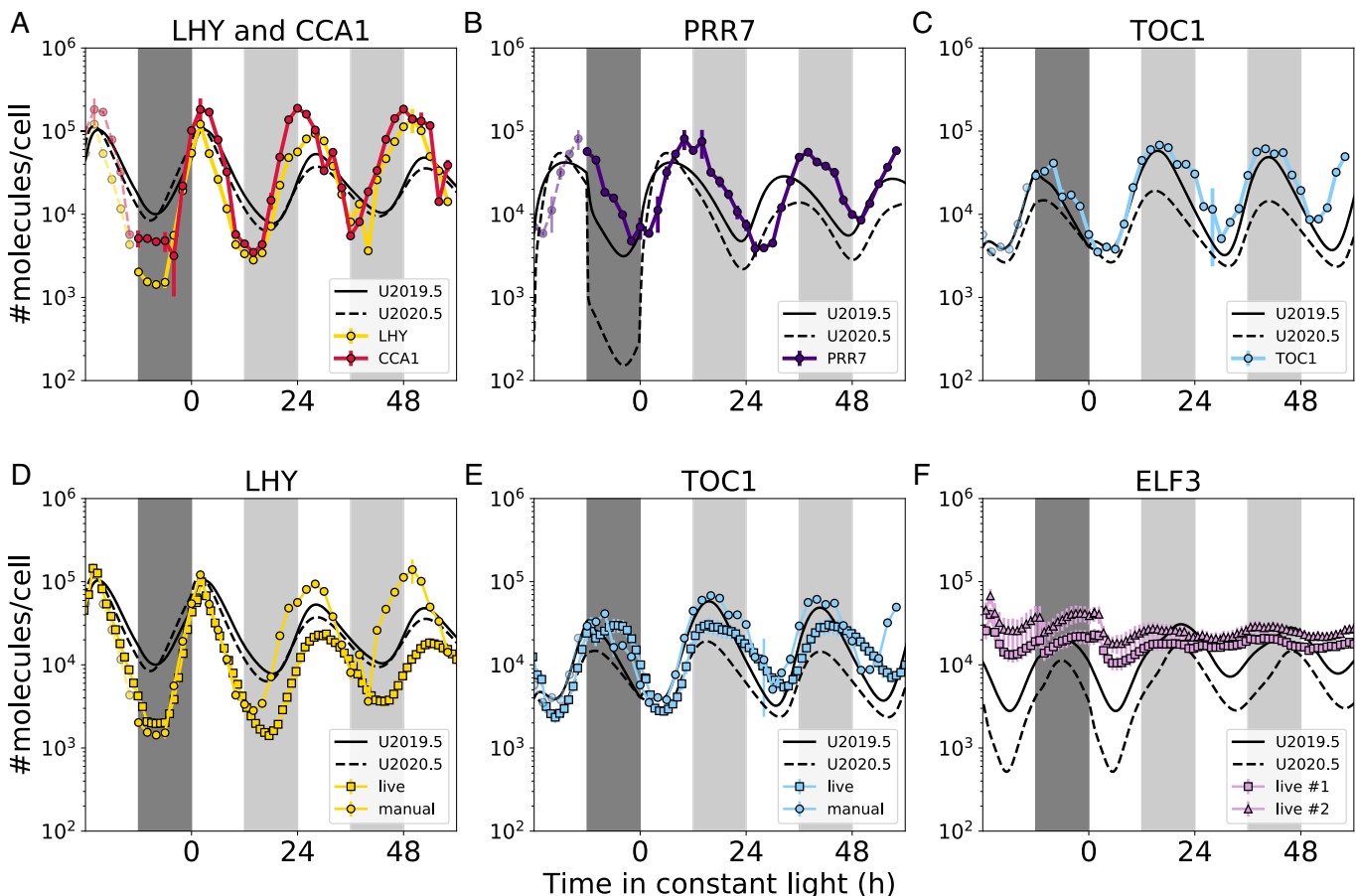

**Figure 6. Absolute protein quantification verifies the mass scale of the clock models.**

(A–C) Reporter protein levels were measured in calibrated NanoLUC assays, for fusions to (A) LHY (yellow) and CCA1 (red), (B) PRR7 and (C) TOC1, in extracts of plants harvested every 2 h from dusk under a 12L:12D cycle followed by 60 h constant light from time 0 h. Measured protein levels closely match the simulated levels from models U2019.5 (solid black line) and U2020.5 (dashed black line). Plants were grown for 21 days in 12L:12D before sampling started. Data from 0 to 10 h are double-plotted at −24 to −14 h (dashed lines). Protein data are means of duplicate biological samples, each of 5 plants, with technical triplicate assays; error bar = 1 SEM. (D–F) In vivo reporter assays suggest ELF3 levels. Seedlings were grown in micro-well plates (4 seedlings per well) for 10 days in 12:12D and furimazine substrate was added to each well. NanoLUC activity was measured hourly, in seedlings carrying LHY (D), TOC1 (E) and ELF3 (F) reporters, under one further 12 L:12D cycle followed by constant light in an automated luminometer. The falling trend due to furimazine decay was removed from the in vivo signals (Fig. EV3). A single scaling value matched in vivo data for LHY (D) and TOC1 (E) to the cognate signals from extracts (A, C). The same scaling factor was applied to the data from two ELF3 reporter lines (F), suggesting the levels of ELF3 protein in vivo. Light interval, white background; dark interval, dark grey shading; anticipated dark interval during constant light, light grey shading.

in the in vitro timeseries, possibly due to its lower time resolution (Fig. 6A). The in vivo assays of the four, well-expressed clock reporter genes showed quantitative consistency, despite some finer features that remain to be characterised, suggesting that the ELF3 reporter could also be characterised in vivo.

The transgenic ELF3 fusion plants expressed the reporter at similar or higher peak signal levels to the other four clock proteins, with a broad peak starting at ZT12 under constant light (Fig. 6F), as expected from the RNA timeseries. The rhythmic amplitude was smaller than for the other clock protein fusions, under 2-fold, consistent with the simple model's prediction and its underlying RNA timeseries data (Fig. 2). These consistent features suggested that the peak ELF3 protein levels were also ~40,000 molecules per cell (Table 1), by comparison to in vivo results for the other four clock proteins. Under LD cycles, the broad peak of ELF3 reporter signal was interrupted by an abrupt, transient signal decline of around 30% from ZT12 and by a minor peak at ZT2. The detailed

dynamics of this reporter system need further characterisation, before such short-term features can be interpreted.

For LUX, the transgenic line tested in vivo showed clearly-rhythmic expression with a peak at ZT12 as expected (Fig. EV1F). The original line's weak activity in vitro was qualitatively consistent with this rhythmic waveform (Fig. EV2C). Calibrating the in vivo data as above for ELF3 suggested a peak of 5000 LUX proteins per cell, a preliminary estimate far below the ~100,000 predicted by the simple model (Table 1) and which awaits confirmation.

The in vivo assay allows rapid testing of multiple environmental conditions, illustrated here by repeating the assays under short- and long-photoperiod conditions, 8L:16D and 16L:8D (Figs. 7C and EV5). The phase of the night-time rise in LHY and CCA1 signal was clearly sensitive to the time of dusk, being delayed under long compared to short photoperiods, as was the waveform of TOC1, as expected (Edwards et al, 2010; Flis et al, 2016). A similar delay in the PRR7 phase was also expected (Flis et al, 2016) though

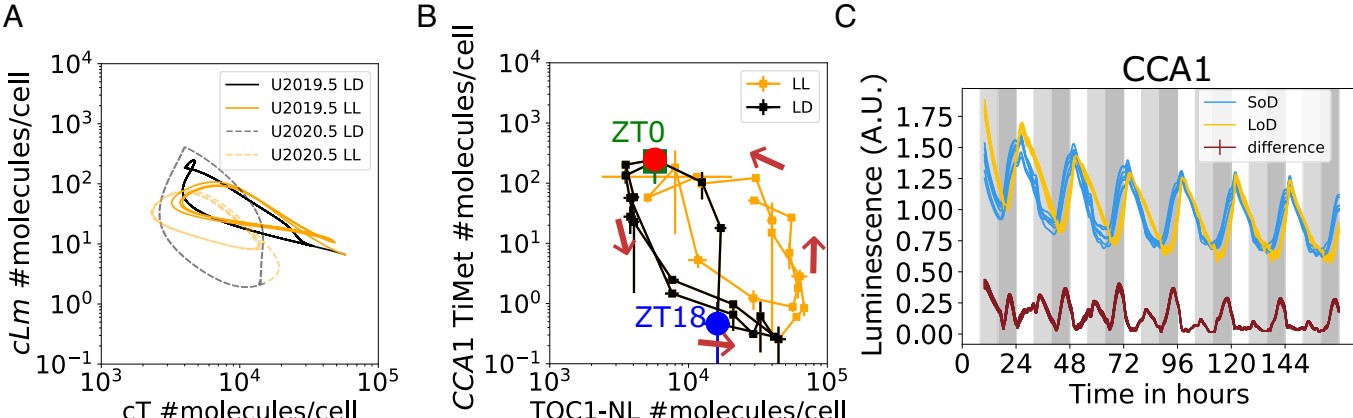

**Figure 7. Dynamics of measured and modelled clock proteins.**

Phase plane diagrams show light-sensitive accumulation of TOC1 protein compared to its target *LHY* mRNA. (**A**) Variables *cT* and *cLm* in models U2019.5 (dashed lines) and U2020.5 (solid lines), and (**B**) TOC1 levels in extracts (from Fig. 6C) and TiMet *LHY* mRNA data (Flis et al, 2015), under 12L:12D cycles (black lines) and constant light (yellow lines). Markers in (**B**) show the first (green) and second (red) ZT0 (lights-on) and the intervening ZT18 (mid-night) under 12L:12D, and the direction of time (arrows). The last data point in black is ZT12 under 12L:12D. In data (**B**), TOC1 levels then fall in 12L:12D but remain high under LL as *LHY* mRNA rises, which is not observed in the models (**A**). Error bars, 1 SEM. (**C**) In vivo recording reflects expected light-responsiveness, under short (8L:16D, cyan lines; SoD) compared to long photoperiods (16L:8D, yellow lines; LoD). Seedlings carrying the LHY protein reporter were grown for 10 of SoD or LoD in a multi-well plate then recorded hourly for 7 days in the same conditions, using an automated luminometer. Data were log-transformed and normalised to the mean of each timeseries (giving arbitrary units, A.U.). Data points for each seedling were connected with a cubic spline interpolation to facilitate comparision despite slight differences in sampling times. The absolute difference between the means (red line) emphasises the earlier rise of expression in the night under 8L:16D. White background, light interval; light grey, dark interval in 8L:16D only; dark grey, dark in both conditions.

photoperiod-responsive changes in the PRR7 reporter profile were less clear (Fig. EV5). Overall, the NanoLUC reporter fusions can report temporal protein profiles in vivo and have potential to indicate absolute protein levels in some cases, by comparison to lines that have been tested by calibrated assays in vitro.

## Discussion

Converting mathematical models of the plant clock from arbitrary to absolute mass units is intended to improve model quality, by harnessing biochemical data (Urquiza-García and Millar, 2021). Absolute units are also helpful for any circuit engineering that includes previously-characterised components. The U2019.5 and U2020.5 models continue this process, by rescaling the previous model versions to match the clock protein levels, which we predicted from absolute RNA levels using the simple model. Protein levels measured in extracts of transgenic plants by the calibrated, NanoLUC reporter fusion technique were consistent with these predictions for LHY, CCA1, PRR7 and TOC1 (Fig. 6, Table 1). The models' simulated $K_d$'s for DNA binding matched calibrated, empirical data from the literature for LUX binding within the EC but not for LHY/CCA1 (Fig. 4, Table 2, Table EV4). The biochemical data that were available in absolute units, or which we could calibrate, were temporarily limiting, so we share the data and models (see Data availability) for other researchers to pursue the remaining discrepancies such as those outlined below and in Dataset EV1.

### Absolute numbers of clock proteins

The success of the simple model suggests that its assumptions, together with the RNA data in absolute units and estimated protein degradation

rates, were sufficient to estimate the dynamic, clock protein levels. In other words, the bulk levels of these clock proteins might be rather simply regulated. The simple model's approach could justifiably be repeated to estimate the levels of other proteins, and extended to test where more complex biochemical mechanisms, such as translational regulation, are functionally significant. Acquiring the necessary RNA data from absolutely-calibrated assays (Flis et al, 2015) could be easier than developing a calibrated, NanoLUC fusion reporter assay in extracts of transgenic plants (Urquiza-García and Millar, 2019). For example, the reporter assays could quickly test protein numbers under different conditions from those reported here, to understand the biochemical mechanisms for the canonical 'temperature compensation' of circadian period in constant conditions (as modelled in Gould et al, 2013) and/or the adaptation to fluctuating, natural conditions (see Future perspectives: models informed by genome sequence, below).

The NanoLUC reporter assays in plant extracts estimated clock protein numbers that vary rhythmically from ~1000 to ~100,000 copies per cell, for LHY, CCA1 and PRR7, with about half that peak level of TOC1. Individual plant cells probably have higher-amplitude protein cycles than these bulk figures. in vivo assays suggested that ELF3 and LUX levels peaked at 40,000 and 5000 protein copies per cell, respectively, though these preliminary estimates await confirmation. Our quantification of plant transcription factor abundance seems compatible with an estimate of 30,000 monomers of phytochrome B, which interacts with transcription factors (~3000 phyB in each of up to 10 photobodies per nucleus, Kim et al, 2023), or ~1 million copies of one of three histone H1 variants (AT2G30620; Probst et al, 2020) out of 160 million proteins in total per mesophyll nucleus (Heinemann et al, 2020). The equivalent ranges for mammalian, fungal and cyanobacterial clock proteins, respectively, include 1000–15,000 copies of PER2 (Narumi et al, 2016; Smyllie et al, 2016), low 10's of

copies of FRQ (Merrow et al, 1997) and 10,000–25,000 copies of KaiB (Chew et al, 2018; Kitayama et al, 2003) per cell or per nucleus. The apparently larger numbers of plant clock proteins might reflect functional requirements other than circadian timing, for example due to the cell biology of plant nuclei, or of these particular proteins and genomic target sites. Copy number data for other plant transcription factors are now required, to determine whether these clock proteins are unusual.

The protein numbers are larger than the 'system sizes' inferred only from the observed variability of Arabidopsis rhythms using stochastic models (Guerriero et al, 2012; Gould et al, 2018; Greenwood et al, 2022), similar to the equivalent comparison for the mammalian clock proteins (Forger and Peskin, 2005; Leise et al, 2012). The large protein numbers and high rhythmic amplitudes are expected to reduce the stochasticity of circadian timing at the cellular level but might limit its resetting by input signals (Pittendrigh et al, 1991), and contribute to the observed reproducibility of clock gene expression waveforms in plants under laboratory (Flis et al, 2015) and natural conditions (Nagano et al, 2012).

## Refining the modelled protein profiles

Our reporter fusion data were not used in model construction. The U2019.5 and U2020.5 model simulations match the timing of these molecular waveforms relatively well, reflecting the fact that the NanoLUC reporter data are similar to past expression studies in arbitrary mass units, which had informed model construction. Three examples illustrate the detailed questions on absolute protein levels, before we address the challenges of DNA binding (see Future perspectives: investigating $K_d$ in vivo).

First, our data allow a new focus on the amplitude of gene expression rhythms, which is functionally important for higher-order circadian properties. The 1000-fold amplitude of CCA1 and LHY RNA expression waveforms under LD cycles (Flis et al, 2015) drives high-amplitude protein rhythms in the simple model but the full models have lower-amplitude protein rhythms, for example 3-fold to 8-fold lower for CCA1 and LHY (Table 1). Two technical factors likely contribute to this difference. The method for testing model parameters does not match the minimum and maximum points in particular, unless specific steps are taken (Troein et al, 2011; Fogelmark and Troein, 2014), in part because focussing on individual data points risks over-fitting. Our models also include some conservative choices that affect amplitude. Their clock genes respond to transcriptional regulators with moderate sensitivity (Hill factors fixed at 2), for example, whereas greater ultra-sensitivity can be observed in other systems. These choices might be relaxed based upon future evidence. Amplitude changes can also be informative, for example comparing the amplitude under LD cycles, where direct light/dark signals contribute, with only endogenous, circadian regulation under constant light (Fig. 6A, Table 1). Balancing the contributions of many, parallel regulators is a common challenge in modelling complicated systems. However, the direction of amplitude change is simulated correctly by U2019.5 and U2020.5 for LHY, CCA1 and PRR7 (falling in LL) and the remaining absolute discrepancies are small, so validation studies with this level of accuracy might prove laborious.

Second, for TOC1, the full models closely match its protein rhythms and correctly increase in amplitude in LL relative to LD conditions (Fig. 6A, Table 1), yet the model's protein levels are not

necessarily comparable with the fusion protein data. The phase plane diagrams (Figs. 7b and EV4) show LHY and CCA1 mRNA levels in LL rise from trough to peak while TOC1 fusion protein levels are at or close to their peak, implying that measured TOC1 is not always an effective repressor of LHY and CCA1 transcription. In the models, levels of these transcripts rise only as TOC1 protein falls (Fig. 7A). The models represent only active TOC1 repressor, because there was no data to inform modelling of inactive TOC1 protein. The phosphorylation of TOC1 and other PRR proteins (Wang et al, 2010), in particular by CK1 (Uehara et al, 2019), might be sufficient to inactivate them at the end of the subjective night, though additional clock components could also contribute. Such phosphorylation might reflect the "phospho-dawn" observed in proteome-wide, rhythmic protein phosphorylation in the green lineage (Noordally et al, 2018; Kay et al, 2021; Krahmer et al, 2022; Noordally et al, 2023).

The full models depart furthest from the data in the third example, of ELF3 under LD cycles. ELF3 RNA amplitude is only around 10-fold (Flis et al, 2015), or 1.7-fold for the ELF3 protein reporter in vivo under LD cycles (Fig. 6B), identical with the amplitude predicted from the ELF3 RNA rhythm by the simple model (Table 1). Other assays also detected significant ELF3 protein at all phases in Arabidopsis, consistent with a modest amplitude (Hicks et al, 2001; Nieto et al, 2022). However, the full models simulated 7- or 20-fold ELF3 protein amplitudes in LD, with 10-fold less protein at the trough than predicted from the RNA (Fig. 3). The full models regulate ELF3 using the observed light regulation of ELF3 stability by the COP1 system (Yu et al, 2008), which the models represent in detail (Pokhilko et al, 2011). ELF3's partner proteins (GI, ELF4 and LUX) are also multiply regulated by light inputs, potentially contributing regulation in the plant that the current models partially ascribe to ELF3. This reinforces the need to quantify the protein partners and their various complexes (Pokhilko et al, 2011), for example by absolutely-calibrated mass spectrometry (Narumi et al, 2016), in timeseries (Krahmer et al, 2019), in order to constrain these, parallel regulatory mechanisms of the Evening Complex.

## Future perspectives: interpreting NanoLUC profiles in vivo

The NanoLUC fusion reporters promise a rich source of quantitative, protein-level timeseries data for the modelling of plant clock gene circuits, for example to understand the interaction of light-regulated translation with rhythmic RNA abundance (Piques et al, 2009; Seaton et al, 2018; Bonnot and Nagel, 2021), as well as the genetic analysis of other regulatory systems. The impact of cellular factors on bioluminescence in vivo, such as variation in the substrate $O_2$ concentration in transgenic plants, remains to be established in detail. Sensitivity to such factors in principle offers the opportunity to report other cellular parameters (Aflalo, 1991; Feord et al, 2019) but also affects the detailed interpretation of in vivo imaging data, such as the rapid fall in fusion reporter signals upon transition to darkness (Figs. 6E,F, EV3, EV5). Studies using nominally-constitutive expression constructs will be required to interpret these details of the waveform, as previously for the firefly luciferase reporter (Millar et al, 1992; Van Leeuwen et al, 2000; Edwards et al, 2010).

## Future perspectives: investigating $K_d$ in vivo

Tackling the model recalibration as a "Fermi problem" (Phillips and Milo, 2009) depends upon many published studies, and

guarantees that many refinements to our approach will be possible (Dataset EV1). The full models' simulated $K_d$ values for the Evening Complex binding to its target genes (0.3–8.6 nM) overlapped the calibrated range of $K_d$'s for the LUX DNA-binding domain in gel-shift assays (1.3–3.2 nM) (Fig. 4, Table 2, Table EV4). No closer agreement could be expected given the approximations involved. This result depends on LUX levels and on the fraction of LUX protein that contributes to bind DNA in the EC, which emerges in the models only from indirect, functional constraints. For example, the EC must act relatively slowly in its negative feedback on the *LUX* and *ELF4* promoters (modelled by slow and incomplete LUX incorporation), in order to set the 17 h period of rhythms in *lhy;cca1* double mutant plants (Locke et al, 2005; Pokhilko et al, 2012; Urquiza-García and Millar, 2021). The equivalent incorporation of some mammalian clock proteins, in contrast, has been directly tested (Aryal et al, 2017; Koch et al, 2022).

Our experimental data supported the simple model's estimates of clock protein levels for LHY and CCA1 (Table 1), which bind DNA directly in the full model. These protein levels simulated $K_d$ values that were 1000-fold higher (lower binding affinity) than the measured $K_d$ values in vitro, extrapolated to the observed CCA1-binding regions (Fig. 4, Table 2). Among many possible contributions to this discrepancy, the in vitro data tested single binding sequences, whereas the total protein number in cells has evolved with many specific binding sites and non-specific binding, titrating the available protein across the entire genome. The number of specific binding sites measured by ChIP-seq is close to 1000 (Adams et al, 2018; Ezer et al, 2017; Kamioka et al, 2016; Nagel et al, 2015). The mathematical form of our clock models assumes that the clock protein pools are large enough that the protein consumed by binding to target sites has negligible effect on the

available protein pools. That assumption could have been contradicted if just a few thousand clock proteins were present per cell, close to the number of binding sites. In that counterfactual case, a higher binding affinity (lower $K_d$ value) would be required in our models to recognise that additional sites were competing to bind the clock protein, though these sites were not explicitly represented in the model. Recalibrating the model's $K_d$ values using the observed number of CCA1 binding sites gave affinity values that overlapped with the in vitro estimates for binding to the same ChIP-seq regions (Fig. 4, Table 2), another remarkable result. However, the simple model and the experimental measures in fact agreed on protein numbers around 100,000 per cell. Sequestering of clock proteins at around a thousand observed ChIP-seq sites is not expected to alter these protein concentrations significantly, consistent with our current models' assumptions. It is possible that the measured, bulk clock protein levels might over-estimate the protein available for promoter binding due to mechanisms absent from the model, such as protein partitioning outside the nucleus (Yakir et al, 2009), protein titration (Buchler and Louis, 2008), clustering of proteins within the nucleus, a processing step akin to the formation of a smaller EC pool from a fraction of the bulk LUX protein or any combination of these mechanisms (Jeong et al, 2022b; Yao et al, 2022). Otherwise, the high DNA-binding affinities of CCA1 and LHY measured in vitro contrast with the lower affinities (higher $K_d$ values) required in the mechanistic models.

The dissociation constant $K_d$ is a ratio of the binding and dissociation rates, $K_{on}$ and $K_{off}$, either of which could be altered in the relevant in vivo conditions compared to the in vitro assays. The binding rate could be reduced if the accessibility of promoter regions is significantly lower in vivo, where nucleosomes and other chromatin or regulatory proteins can limit binding rate compared to naked DNA

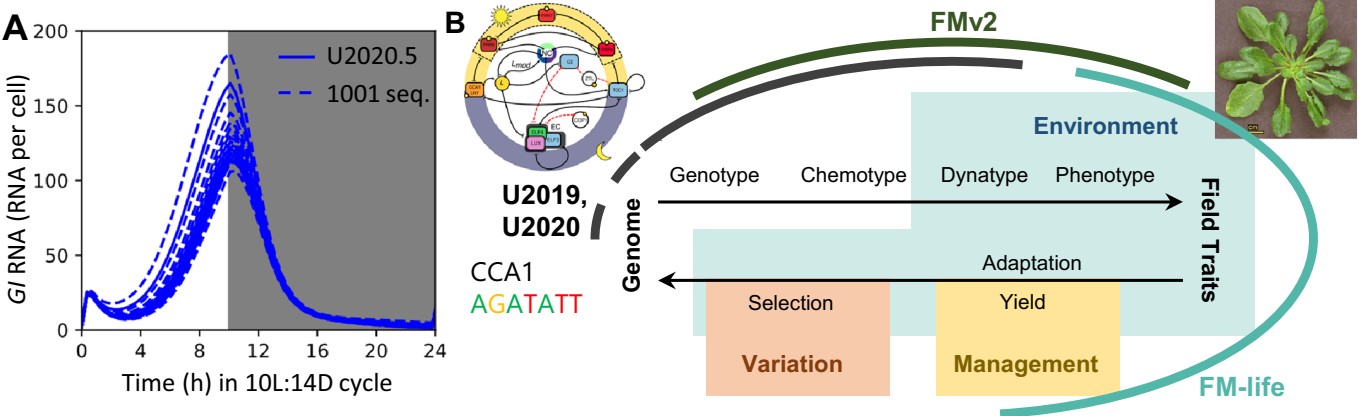

**Figure 8. Connecting genome sequence variation to phenotypic effects across scales.**

Natural genetic variation in promoter sequences is predicted to alter the molecular phenotype (dynatype). (**A**) *GI* transcript levels under 10L:14D cycles were simulated in U2020.5 (white region, light interval; shaded region, dark interval). The $K_d$ for CCA1 binding to the promoter of *GI* was calculated using the EMA matrix (Fig. 5), for all *GI* promoter sequences from the 1001 Genomes Project (dashed lines). The model retained the default *GI* gene (solid line, second highest peak), while a second copy that simulated only *GI* RNA production was tested with the $K_d$ for each promoter sequence and plotted (dashed lines). The range of dynamics shown reflects only altered *GI* transcription rates, without the effects of altered GI protein dynamics. (**B**) Connecting models (adapted from Millar, 2016 Fig. 4c). The central circuit represents the conceptual steps as a genome, in cells, builds organismal traits (upper arrow) in a given environment (green shading). Those traits and potentially management inputs (yellow), in populations, lead to selection on genome sequences (lower arrow). The black, green and cyan arcs represent, respectively, the clock gene circuit models such as U2020 (schema from Urquiza-García and Millar, 2021 Supp Fig. 3b), the Framework Model version 2 for clock-controlled seedling and rosette growth in simple environments (FMv2; Chew et al, 2022), and the FM-life model for whole-lifecycle simulation under natural environments (Zardilis et al, 2019). This paper illustrates how a clock gene circuit model can incorporate genome sequence data (dashed black arc), *via* promoter sequences that alter CCA1 binding, as in (**A**). Others might connect such models in future and add simulated genetic variation (pink), in order to explain and predict both the operation and the evolution of the plant clock genes.

in vitro. ChIP-seq from intact chromatin (Adams et al, 2018) revealed many fewer LHY binding sites (around 700 sites) than the binding potential of LHY from DAP-seq on naked genomic DNA (around 18,000 sites) (O'Malley et al, 2016), for example. The dissociation rate could also be increased in vivo if the minority of bound proteins are removed from the promoter by nucleosomes or by an active degradation process (Spoel et al, 2009). Quantifying these biochemical mechanisms in planta is technically challenging. Our models should help to understand their effects at larger scales.

## Future perspectives: models informed by genome sequence

The absence of measured $K_d$s for clock protein binding to target promoter sequences led us to estimate these affinities from protein-binding microarray data (Fig. 5), an approach that should be applicable to other transcription factors, pending direct assays of binding in vivo. Our results analyse the CCA1 and EC binding sequences identified by ChIP in any of the model's target genes (Table 2, Table EV4) but the approach applies equally to any sequence variant. Figure 8A shows the predicted effect of observed, natural variation in the CCA1-binding sequences in the promoter region of *GIGANTEA (GI)* among Arabidopsis accessions sequenced by the 1001 Genomes Project (Weigel and Mott, 2009) (Data ref: 1001 Genomes Project). Any sequence-driven analysis that informs a parameter in the full model is within the scope of this approach, so a growing number of genome sequence variants might become interpretable. Alternative RNA splicing is a sequence-dependent regulatory mechanism (James et al, 2018), for example, which underlies the control of LHY translation that

forms part of the plant clock's response to temperature (Gould et al, 2013). For Arabidopsis, it seems premature to expect that DNA-binding affinities might be predictable from sequence variants in the DNA-binding domains of clock proteins, informed by biophysical data and sequence-base protein structure predictions (Jumper et al, 2021), but this might be tractable in other systems (Narasimamurthy et al, 2018).

Sequence-informed, mechanistic models such as U2019.5 and U2020.5 promise to link these data on genetic biodiversity quantitatively to their effects at larger scales. *GI* expression both contributes to the photoperiodic control of flowering among *A. thaliana* accessions (de Montaigu et al, 2015) and has known and modelled molecular mechanisms (Seaton et al, 2015). The waveforms of Arabidopsis clock gene expression are already linked to whole-organism traits in the Framework Models (Kinmonth-Schultz et al, 2019; Chew et al, 2022). An initial extension to ecological data shows the potential to understand evolutionary adaptation in mechanistic terms (Burghardt et al, 2015; Zardilis et al, 2019). In the context of the Framework Models (Fig. 8B) such as our FMv2 and FM-Life, a sequence-informed clock model might predict the change in flowering time in particular environmental conditions, and hence the phenology of flowering in nature. If population genetics (genetic variation and selection) can be incorporated into this system of models in the future, then plant science might make sense of the plant circadian system quantitatively and mechanistically, in the light of evolution (Dobzhansky, 1973; Millar, 2016).

## Methods

**Reagents and tools table**

| Reagent/Resource | Reference or Source | Identifier or Catalog Number |
|---|---|---|
| **Experimental models** | | |
| Col-0 (*A. thaliana*) | NASC | Id. 30210 |
| Col-0 CCA1::LUC | Kamioka et al (2016) | N/A |
| Col-0 cca1-1/lhy-11 CCA1::LUC+ | Kamioka et al (2016) | N/A |
| Col-0 cca1-1/lhy-11 CCA1::LUC+ CCA1p:CCA1-NL3F10H #63.1 | This study | N/A |
| Col-0 cca1-1/lhy-11 CCA1::LUC+ LHYp:LHY-NL3F10H #35.1 | This study | N/A |
| Col CCR2:LUC+ | Farré et al (2005) | N/A |
| Col prr7-3/prr9-1 CCR2::LUC | Farré et al (2005) | N/A |
| Col prr7-3/prr9-1 CCR2::LUC+ PRR7p:PRR7-NL3F10H #18.2 | This study | N/A |
| Col toc1-2 CCA1:LUC+ | Gendron et al (2012) | N/A |
| Col toc1-2 CCA1:LUC+ TOC1p:TOC1-NL3F10H | This study | N/A |
| Col lux-2 CAB2:LUC+ | Hazen et al (2005) | N/A |
| Col lux-2 CAB2:LUC+ LUXp:LUX-NL3F10H #21.2 | This study | N/A |
| Col lux-2 CAB2:LUC+ LUXp:LUX-NL3F10H #9.2 | This study | N/A |
| Col elf3-2 | Nusinow et al (2011) | N/A |
| Col elf3-2 ELF3p:ELF3-NL3F10H #6.1 | This study | N/A |
| Col elf3-2 ELF3p:ELF3-NL3F10H #12.1 | This study | N/A |
| Col elf3-2 ELF3p:ELF3-NL3F10H #65.1 | This study | N/A |
| Rossetta 2(DE3)pLysS Competent Cells - Novagen | Sigma | Cat. #71403-3 |

| Reagent/Resource | Reference or Source | Identifier or Catalog Number |
|---|---|---|
| ABI strain (*Agrobacterium tumefaciense*) | Prof. Seth Davis, York University, UK | N/A |
| **Recombinant DNA** | | |
| pGWB635 | Nakamura et al (2010) | N/A |
| pET28a(+)::MBP-NanoLUC-3xFLAG-10xHis | Urquiza-Garcia and Millar (2019) | Addgene #141291 |
| pGWB601NL3F10H (BASTA^R) | Urquiza-Garcia and Millar (2019) | Addgene #141287 |
| pDONR221:CCA1pro::CCA1 | This study | N/A |
| pDONR221:LHYpro::LHY | This study | N/A |
| pDONR221:PRR7pro::PRR7 | This study | N/A |
| pDONR221:TOC1pro::TOC1 | This study | N/A |
| pDONR221:LUXpro::LUX | This study | N/A |
| pGWB601:CCA1pro::CCA1-NL3F10H | This study | N/A |
| pGWB635:CCA1pro::CCA1-LUC+ | This study | N/A |
| pGWB601:LHYpro::LHY-NL3F10H | This study | N/A |
| pGWB635:LHYpro::LHY-LUC+ | This study | N/A |
| pGWB601:PRR7pro::PRR7-NL3F10H | This study | N/A |
| pGWB635:PRR7pro::PRR7-LUC+ | This study | N/A |
| pGWB601:TOC1pro::TOC1-NL3F10H | This study | N/A |
| pGWB635:TOC1pro::TOC1-LUC+ | This study | N/A |
| pGWB601:ELF3pro::ELF3-NL3F10H | This study | N/A |
| **Oligonucleotides and other sequence-based reagents** | | |
| PCR primers | This study | Table EV6 |
| **Chemicals, Enzymes and other reagents** | | |
| Agar | Sigma | Cat. #A1296 |
| ½ Muraschige and Skoog salts | Duchefa | Cat. #M0222.0001 |
| MES | Sigma | Cat. #69892 |
| IPTG | Sigma | Cat. #I6758 |
| Ni-NTA Agarose 25 ml | Qiagen | Cat. #30210 |
| PMSF | Merck | Cat. #P7626 |
| Protease Inhibitor Cocktail | Sigma | Cat. #9599 |
| MG132 (S)-MG132 | STEM CELL Technologies | Cat. #73264 |
| Imidazol | Sigma | Cat. #I5513 |
| EDTA | Sigma | Cat. #E4884 |
| EGTA | Sigma | Cat. #324626 |
| NanoGlo® Luciferase Assay | Promega | Cat. #N1110 |
| D-Luciferin (potassium salt) | Biosynth | Cat. #FL08607 |
| Acetosyringone | Sigma | Cat. #D134406 |
| DMSO | Sigma | Cat. #D8418 |
| Glufosinate-ammonium | Sigma | Cat. #45520 |
| Silwet® L-77 | Plant media | SKU# 30630216 |
| Sucrose | Sigma | Cat. #S0389 |
| **Software** | | |
| Docker Engine | Docker Inc. | https://www.docker.com/ |
| uurquiza/urquiza2019a_tellurium_sloppycell | This study, in dockerhub | docker pull uurquiza/urquiza2019a_tellurium_sloppycell |
| ImageJ64 | NIH, USA | http://imagej.nih.gov/ij |
| Wasabi (OrcaII controlling software) | Hamamatsu Photonics, Japan | N/A |

| Reagent/Resource | Reference or Source | Identifier or Catalog Number |
|---|---|---|
| Biodare2 | Zielinski et al (2022) | https://biodare2.ed.ac.uk/ |
| **Other** | | |
| Tristar² LB 942 microplate reader | Berthold | N/A |
| EMCCD OrcaII | Hamamatsu Photonics, Japan | model C4742-98 |
| Tissue Lyser | Qiagen | |
| 96-well black sterile plates | LUMITRAC, Greiner | Cat. #655075 |
| TopSeal-A plus plate seal | Perkin Elmer | Cat. #6050185 |
| Vibra-cell sonicator | Sonics & Materials Inc., Newton, Connecticut, USA | N/A |

## Computational procedures

Analysis of published data to estimate translation and protein degradation rates for the simple model and analysis of protein-binding microarray data using Error-Model Averaging are described in the Appendix. All the mathematical models were written in the human-readable language Antinomy and transformed into SBML files using the python package Tellurium (Choi et al, 2018). Rescaling parameters were introduced into the models U2019.3 and U2020.3 (Urquiza-García and Millar, 2021) to match protein levels to the levels predicted by the simple model, creating U2019.4 and U2020.4, respectively (Fig. 3). The scaling factors were estimated using the least-squares method implemented by the minimize function of the python package lmfit (Newville et al, 2023), with a custom cost function that generates numerical solutions of the model, and also plots simulation results, using the Tellurium software (Choi et al, 2018). Unless modelled otherwise, clock proteins were assumed to be nuclear and their concentrations were calculated using a published nuclear volume (Tirichine et al, 2009) from our previous parameter compendium (Millar et al, 2019). The modelling software system was run in a Docker container, described by the Docker file shared in the data compendium (see Data availability).

## Cloning and prototyping

The Arabidopsis Col-0 TAIR10 genome assembly was used a reference genome. The genomic region of clock genes listed in Table EV5 was amplified from Col-0 genomic DNA with primers described in Table EV6, apart from *ELF3*. The fragments were then recombined into pDONR221 using BP II clonase (Invitrogen) and sequence determined by Sanger sequencing (Genepool, Edinburgh Genomics, Edinburgh, UK). The genomic regions were recombined into pGWB601:NanoLUC-3FLAG-10His using LR clonase, as described (Urquiza-García and Millar, 2019). *ELF3* was cloned in the same destination vector using Gibson cloning with primers described in Table EV6. The constructs were then transformed into *E. coli* DH5α and selected on Spectinomycin. The candidate plasmids were tested for NanoLUC activity and rhythmicity (Appendix Fig. S3) by transfecting protoplasts as described in Appendix (Hansen and van Ooijen 2016; Urquiza-García and Millar, 2019).

## Plant transformation

The pGWB601:XpX-NL3F10H vectors carrying genomic regions of interest (X in the plasmid name) were transformed using liquid nitrogen into *Agrobacterium tumefaciens* ABI (kindly provided by Prof. Seth Davis, York). Arabidopsis plants were transformed by floral dipping (Wang, 2015). Primary transformants were selected for 3:1 segregation of BASTA resistant:sensitive progeny, to identify single-insertion lines. Single-insertion transgenic lines were analysed for phenotypic complementation in the T3 generation. For each construct, 15 or more homozygous lines with segregation consistent with single-insertion events were tested for a circadian period closest to the wild-type control and, for LUX and ELF3, also for hypocotyl elongation (Appendix; Fig. EV1).

## In vivo luciferase imaging

Period determination of transgenic lines was performed by in vivo luciferase reporter gene imaging, essentially as in Southern et al (2006), see Appendix. Period analysis with the FFT-NLLS algorithm in the public Biodare2 resource (Zielinski et al, 2014, 2022) used data starting after the first day in constant light conditions. Transgenic lines with the closest period and relative amplitude error relative to the corresponding WT control lines were retained for further analysis.

In vivo luminometry of NanoLUC reporters was similar to (Urquiza-García and Millar, 2019). Four sterilised seed of each homozygous transgenic line were sown per well of a white, flat-bottomed 96-well plate that contained 150 μl of solid agar media (see Appendix). After one day at 4 °C, plates were incubated for 10 days in 12L:12D conditions and treated with 50 μl of 1:50 of furimazine substrate solution (Promega, Southampton UK): 0.01% Triton X-100. The plate containing 8 biological replicates per construct was then assayed by luminometry every 30 or 60 min, in an automated TriStar2 S LB 942 luminometer (Berthold Technologies, Harpenden, UK) at 21 °C, and exposed to monochromatic blue and red LED light with a total 50 μmol m$^{-2}$s$^{-1}$, 12L:12D photoperiod between readings for three days, followed by constant light. The falling trend in the signal of all reporters over several days in constant light is assumed to result from decay of the substrate furimazine. In vivo data from TOC1 and CCA1 reporter plants were scaled such that the means matched the cognate in vitro data. This scaling was repeated on detrended in vivo data for ELF3 and LUX reporters, to estimate their protein copy numbers.

## Calibration of the NanoLUC assay in plant extracts

Briefly (as a step-by-step protocol is provided below), protein produced from the MBP-NanoLUC-3FLAG-10His construct in *E. coli* was purified as previously described (Urquiza-García and

Millar, 2019) and quantified by a linear version of the Bradford protein assay (Ernst and Zor, 2010). Col-0 Arabidopsis plants were grown in 5 cm diameter tissue culture dishes, using media and growth conditions as above, for 10–14 days and collected in 0.1 gramme fresh weight (gFW) aliquots in 2 ml microcentrifuge tubes (Safelock®, Eppendorf), along with two, 2 mm stainless-steel grinding balls. Sufficient MBP-NanoLUC3F10H protein was added to the tissue samples to correspond to starting levels of 0, $1 \times 10^2$, $1 \times 10^3$, $1 \times 10^4$, $1 \times 10^5$, $1 \times 10^6$ monomers cell$^{-1}$, using the previously-measured average of 25 million cells/gFW of leaf tissue (Flis et al, 2015). The aliquots were then frozen in liquid nitrogen and extracted as detailed below, for NanoLUC assays of transgenic plant tissue. A calibration curve was generated on each 96-well plate measured, using 4 biological replicates per NanoLUC dilution. The data were ln-transformed, and a linear regression to these data yielded the calibration standard for each plate.

## Generation of NanoLUC timeseries from plant extracts

Briefly (as a step-by-step protocol is provided below), seed of transgenic lines were sown on solid media as described above. Growth and harvesting conditions matched the TiMet protocol (Flis et al, 2015). After two weeks under a 12L:12D photoperiod of cool white fluorescent light at 21 °C, robust young plants were transferred to F2 + S Levington compost (Frimley, UK) and grown until they were 21 days old. Leaf rosettes were pooled from 5 plants per biological replicate in 2 ml microcentrifuge tubes (Safelock®, Eppendorf) containing two, 2 mm stainless steel grinding balls and flash-frozen in liquid nitrogen. The sample tubes were maintained under liquid nitrogen, ground twice using a Tissuelyser (Qiagen Ltd., Manchester), and returned to liquid nitrogen before extraction. On ice, 150 µl of BSII buffer (100 mM sodium phosphate, pH 8.0, 150 mM NaCl, 5 mM EDTA, 5 mM EGTA, 0.1% Triton X-100, 1 mM PMSF, 1× protease inhibitor mixture (Roche, Basel, Switzerland) and 5 µM MG132 (Stem Cell Technologies, Cambridge, UK)) was added similar to (Huang et al, 2016) but without phosphatase inhibitors. The tubes were thoroughly vortex-mixed, returned to ice, while each tube's mass was individually measured which then was adjusted with BSII buffer to reach a tissue concentration of 0.4 gFw ml$^{-1}$ as recommended by the manufacturer for the protease inhibitors (Sigma-Aldrich). The extracts were clarified by centrifugation at $20,000 \times g$ for 10 min. 20 µl of plant extract per well was added to 96-well plates that were pre-loaded with 80 µl of BI assay buffer (50 mM NaH$_2$PO$_4$, 0.3 M NaCl, pH 8.0 NaOH adjusted). 100 µl of 1:50 Furimazine:NanoGlow (Promega, Southampton UK) were added using a multipipette. After 10 min incubation at 21 °C in a temperature-controlled growth room, the bioluminescence was measured in a Tristar2 S LB 942 luminometer (Berthold Technologies, Harpenden, UK) at 21 °C with a signal integration time of 1.5 s. The number of NanoLUC molecules cell$^{-1}$ was calculated from the bioluminescence values, based the calibration standard (see above). No blinding was performed in these studies.

## Step by step protocol for absolute quantification of protein abundance using *Arabidopsis thaliana* transgenic lines carrying NanoLUC-tagged genes

The full protocol is available on the protocols.io resource, please see Data availability.

### Materials

Equipment.

1. 250 ml Erlenmeyer flasks and shaker (e.g. Innova 44 New Brunswick)
2. High speed centrifuge up to $20,000 \times g$ for 2 ml polypropylene tubes
3. 37 °C orbital incubator shaker
4. Tristar$^2$ LB 942 microplate reader (Berthold)
5. Liquid nitrogen
6. Spectrophotometer (600 nm)
7. Vibra-cell sonicator (Sonics & Materials Inc., Newton, Connecticut, USA)
8. 1.5 ml polypropylene microtubes
9. 2 ml polypropylene Safelock (Eppendorf)
10. 50 ml conical polypropylene centrifuge tubes
11. 2 mm grinding balls
12. 96-well sterile plates (LUMITRAC, Greiner Cat. #655075)
13. Light source (e.g. Light DNA8 Valoya, cool white lamps, LED panels)
14. TopSeal-A plus plate seal (Perkin Elmer Cat. #6050185)
15. Tissue Lyser (Qiagen)
16. Analytical scale

ROBUST media. Weigh Agar (Sigma Cat. #A1296), ½ MS salts including vitamins (Duchefa Cat. #M0222.0001), MES (Sigma). For 400 ml of Media add 4.8 g of Agar and 0.86 g of ½ MS salts with vitamins. Add 300 ml of deionised water and measure the pH. Adjust to 5.5 using either NaOH or HCl (see Note 1). Transfer the pH adjusted solution into a measuring cylinder and fill to 400 ml. Then transfer to a 500 ml borosilicate bottle and sterilize.

NanoLUC expression.

1. 1 M IPTG, dissolve 2.383 g of IPTG (Sigma Cat. #I6758) in 5 ml ddH$_2$O then fill to 10 ml. Prepare 250 µl aliquots in 1.5 ml polypropylene tubes and store at −20 °C.
2. Rossetta 2(DE3)pLysS Competent Cells - Novagen (Sigma Cat. #71403-3). Follow manufacturer instructions for handling and plasmid transformation.
3. Plasmids: pET28a(+) and pET28a(+)::MBP-NanoLUC-3xFLAG-10xHis (Addgene ID 141291) both Kan$^R$ Cm$^R$ have been described before (Urquiza-Garcia and Millar, 2019).
4. Ni-NTA Agarose 25 ml (Qiagen Cat. #30210).
5. Lysis Buffer (50 mM NaH$_2$PO$_4$, 300 mM NaCl, 10 mM Imidazole, pH 8.0 NaOH adjusted).
6. Washing Buffer (50 mM NaH$_2$PO$_4$, 300 mM NaCl, 20 mM Imidazole, pH 8.0 NaOH adjusted).
7. BSIII buffer preparation buffer: 100 mm sodium phosphate, pH 8.0, 150 mm NaCl, 5 mm EDTA, 5 mm EGTA, 0.1% Triton X-100, 1 mM PMSF, Protease inhibitor Cocktail (Sigma Cat. #P9599), and 5 µM MG132 (S)-MG132 (STEMCELL Technologies Cat. #73264).
8. 2 ml Safe Lock tubes (Eppendorf) with $2 \times 2$ mm stainless steel grinding balls. Weigh each individual tube to control for the experimental error associated with weight variations due to the manufacturing process. Label each tube for the specific time point on the side of the tube.
9. NanoGlo® Luciferase Assay (Promega Cat. #N1110).

Plant strains, growth and transformation material.

1. Col-0 (NASC id. 30210) and Col-0 pGWB601::35S:NanoLUC-3xFLAG-10xHis described in (Urquiza-Garcia and Millar, 2019).

2. Binary vectors pGWB401NL3F10H (Km[R], Addgene ID 141285), pGWB501NL3F10H (Hyg[R], Addgene ID 141286), pGWB601NL3F10H (BASTA[R], Addgene ID 141287), pGWB701NL3F10H (Tun[R], Addgene ID 141288).

3. Agrobacterium strains ABI (Kan[R], Cm[R]), AGL-1 (Rif[R], Amp[R]), GV3101 (pMP90) (Rif[R], Gent[R], Km[R]),

4. 300 mM Acetosyringone stock solution (Sigma Cat. #D134406). Weigh 0.588 g dissolve in 5 ml and fill to 10 ml with DMSO (Sigma Cat. #D8418). Can be stored at $-20\,°C$, however, it should be preferably prepared fresh.

5. Infiltration media (½ MS media containing 5% Sucrose, 3 mM MES, pH 5.5 KOH adjusted, 200 µl/L Silwet L-77, 150 µM Acetosyringone) (all Sigma, except MS). For 1 L: 2.165 g ½ MS, 50 g sucrose, 0.59 g 3 mM MES, 200 µl Silwet L-77 and 500 µl of 300 mM acetosyringone stock solution.

6. 2.5 mg/L glufosinate-ammonium (Sigma Cat. #45520), 0.01% Triton X-100 (Sigma Cat. #X100).

7. Hand-pumped sprayer bottle.

8. Levington Advance Seed & Modular F2S Compost mix or equivalent.

9. Pots, Growth Chambers (e.g. Binder KBWF720).

## Method

Exploratory in planta time series (optional). Before performing a discrete time series (manual sampling), we recommend performing a pilot time series experiment using a luminometer to perform whole plant assays. This will provide valuable information about the overall dynamics of the NanoLUC tagged protein. With this information the user can then select the most appropriate time points for absolute quantification assays. In particular, the user is interested in determining the dynamic range of the signal, to select time points of lowest and highest signal level, which can be used to avoid saturation of plate reader readings. Also, these in planta preliminary experiments need only 50 µl of 1:10 Furimazine:0.01% Triton X-100 for each replicate and provide substantial amounts of quantitative data in arbitrary units (see Urquiza-Garcia and Millar, 2019).

1. Melt ROBUST media and then pipette 100 µl into each well that will contain a plant.

2. Once it has cooled down, place 1–3 seeds with a pipette in each well, keeping the number of seeds constant in all wells.

3. Seal the plate with a TopSeal-A plate cover and wrap them with aluminium foil. Stratify at $4\,°C$ for 3–4 days.

4. Germination synchronization can be optimized by giving a 2 h white light pulse (100 µmol m$^{-2}$ s$^{-1}$) followed by 22 h of darkness at $21\,°C$.

5. After 22 h transfer the plate to the growth conditions of interest.

6. Add 50 µl of 1:5 Furimazine:0.1% Triton X-100 on the top of the seedlings of each well.

7. Place the plate in the luminometer and set your desired sampling conditions. For example, for the Tristar LB 942, we recommend measurements every 30 min–1 h with 1.5 s of integration time and 1 min delay in darkness prior to the measurement (see Note 5). Under these conditions we have been able to collect data for up to 9 days in a combination of photoperiodic and constant white light conditions.

Recombinant expression and purification of MBP-Nanoluc-3xFLAG-10xHis (MBP-NL3F10H). Usually this step will be performed just before the absolute quantification will take place, to provide a fresh batch of pure NanoLUC. This step should be started two days before preparing the calibration curve to determine the absolute amount of NanoLUC tagged protein of interest (see section Time series sampling).

1. Perform transformation of chemical competent Rosetta™ 2 (DE3) pLysS cells using pET28a(+) as (Control) and pET28a(+):MBP-NanoLUC-3F10H (or using other transformation method of choice), select on LB Cm$^{34}$ Kan$^{50}$ plate and incubate at $37\,°C$ overnight.

2. Inoculate 50 ml of LB Cm$^{34}$ Kan$^{50}$ with two large colonies and track O.D.$_{600}$. When cell culture reaches an O.D.$_{600}$ = of 0.1, add IPTG to a final concentration of 1 mM, and incubate at $37\,°C$ 200 rpm overnight (see Note 2).

3. Collect the cells by centrifugation at $2400 \times g$ for 15 min and resuspend in Lysis Buffer (2–5 ml/g of wet weight) (see Note 3). Cells can be flash frozen at this point and kept at $-80\,°C$ (see Note 4) or further processed.

4. Lyse cells by sonication using a Vibracell Sonicator or any other accessible sonicator (the conditions need to be optimised for each lab, a curve of protein release as a function of cycles can be created to follow the cell lysis). Transfer 2 ml of lysate to 2 mL Safe lock tubes (2x). Then remove insoluble debris and large particles by centrifugation at $20,000 \times g$ for 20 min at $4\,°C$.

5. Transfer carefully 1.8 ml each cleared lysates (supernatant) to a new 2 ml Safe Lock tube.

6. Add 200 µl of Ni-NTA agarose beads (Qiagen) and incubate at $4\,°C$ for one hour with gentle and continuous inversion. This can be done in a $4\,°C$ room on a rotatory shaker set at 200 rpm.

7. Wash 3 times the Ni-NTA agarose beads with 1 ml Washing Buffer making sure that the agarose beads are completely resuspended.

8. Verify the purity and homogeneity of the purification by SDS-PAGE.

9. Perform protein quantification by a method of choice, in our case we have used a linearized version of the Bradford assay.

Tissue preparation for calibration curve for determining number of molecules per cell. 1. Two weeks before performing the absolute quantification measurements, sterilise Col-0 seeds and plate them in ROBUST media petri dishes. Stratify the seeds for 3–4 days at $4\,°C$ and darkness.

2. Synchronise germination by treating the plates with a 2-h pulse of 100 µmol m$^{-2}$ s$^{-1}$ white light at $22\,°C$, followed by a 22 h period of darkness at $22\,°C$. Then transfer to the growth conditions under which the transgenic plants tagged with NanoLUC will be analysed.

3. Prepare 12 of 2 ml safe-lock Eppendorf tubes by adding 2 stainless steel 2 mm grinding balls on each. Label the tubes. Afterwards, weigh the tubes containing the grinding balls and record the weights on an Excel sheet.

4. Collect 2-week-old plants (total tissue of 100 mg) in a 2 ml SafeLock Eppendorf tube with 2 stainless steel 2 mm grinding milling balls.

5. Create a standard curve by adding purified MBP-NanoLUC-3xFLAG-10xHis (see Note 6). MBP-NL3F10 has a molecular weight

of 66.833 kDa. Assuming 25 million cells/gFW the final concentration of tissue in the BSIII buffer should be 0.4 gFW/ml resulting in a total of 10 million cells/ml. Therefore standard curves with 0, $1 \times 10^2$, $1 \times 10^3$, $1 \times 10^4$, $1 \times 10^5$ and $1 \times 10^6$ molecules/cell can be prepared by adding the corresponding volumes from pure and quantified MBP-NanoLUC-3xFLAG-10xHis. The pure enzyme preparation is added before flash freezing the tissue and performing tissuelyser disruption in order to simulate the possible impact of sample preparation on NanoLUC activity.

6. Flash-freeze the tissue in liquid nitrogen at the beginning of the time-series sampling. Maintain these aliquotes in liquid nitrogen throughout the duration of the time-series sampling (next section).

Plant growth for time series.

1. Sterilise seeds of NanoLUC tagged lines and plate them in ROBUST media petri dishes. Stratify the seeds for 3–4 days at 4 °C and darkness.
2. Synchronise germination by treating the plates with a 2-h pulse of 100 μmol m$^{-2}$ s$^{-1}$ white light at 22 °C, followed by a 22 h period of darkness at 22 °C. Then transfer to the growth conditions under which the transgenic plants tagged with NanoLUC will be analysed.
3. After one week in these conditions transfer healthy plants to compost mix and continue under the experimental conditions of interest until sampling will take place (section Plate reading measurements of NanoLUC activity). If plants are going to be analysed for a longer period of time in vitro growth needs to be optimised by the user.

Time series sampling.

1. Before starting sampling, prepare the required amount of 2 ml safelock polypropylene tubes. Add two 3 mm stainless steel grinding balls. Label the tubes, measure the weight and record this in an excel table for determining the amount of gFW collected.
2. Pool five 21-day-old plants into a single 2 ml safelock polypropylene tube that contains two stainless steel 3 mm grinding balls previously weighed. Flash freeze the sample in liquid nitrogen. Keep the sample stored in liquid nitrogen for the full duration of the sampling and immediately that contains grinding balls. Flash freeze them in liquid nitrogen. Keep them stored in liquid nitrogen throughout the duration of the sampling (see Note 4).
3. After sampling has been completed, perform two rounds of grinding using the Tissue Lyser (Qiagen) with the following settings: 30 hz for 1 min. Perform the same procedure to the calibration curve aliquots.
4. Take one sample at time and record weight in the excel sheet and add 150 μl of ice-cold BSIII buffer for blocking protein degradation (see note 7). Vortex the tubes vigorously and place them on ice.
5. Using the excel sheet determine the tissue mass by subtracting the originally recorded empty tube weight.
6. Calculate the amount of additional ice-cooled BSIII buffer required for a final "concentration" of 0.4 g FW/ml. After adding the additional BSIII vortex thoroughly.
7. Centrifuge the samples at $20,000 \times g$ for 5 min and transfer the cleared crude lysate to ice-cooled 1.5 ml polypropylene tubes. Keep

the tubes in ice for reducing protein degradation and proceed quickly to determine NanoLUC activity.

Plate reading measurements of NanoLUC activity. We recommend the plate layout set up presented in Fig. 9.

1. Calculate the amount of reconstituted NanoGlo® buffer aiming to deliver 100 μl/well. Choose a conical polypropylene tube adequate for the required number of samples.
2. Reconstitute the NanoGlo® buffer by mixing 1:50 Furimazine:NanoGlo®, then pour the substrate in a plate for loading using a multi-channel pipette.
3. Load 80 μl of ice cold BSIII in a 96-well flat bottom black Lumitract Greiner plate (see Note 8). Then proceed to load 20 μl of cleared crude lysate. Keep the plate on ice to reduce protein degradation. This will also reduce sample evaporation.
4. Take 100 μl of reconstituted substrate using a multi-channel pipette and add it to each well. Mix carefully, avoiding the formation of bubbles or foam (see Note 9).
5. Allow the plate to equilibrate to room temperature for 5 min (25 °C) and then proceed to perform the measurements in a plate reader (e.g. Tristar (Berthold) using 1 s integration time).

Data analysis. Using Excel or other similar calculating spreadsheet software. The dominant experimental error can be considered log-normal. Therefore, take the natural logarithm of the data and perform a linear regression against the calibration curve for determining copies/cell from the plate reader measurements.

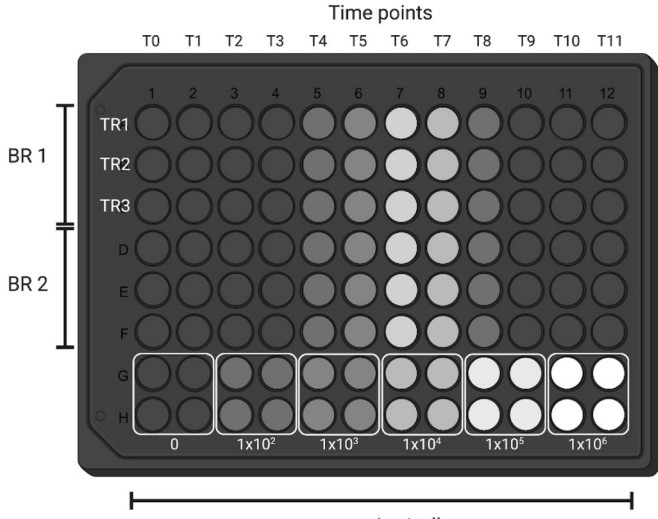

**Figure 9. 96-well black plate layout for time series absolute quantification.**

Time points of series (T0 ... T11). 2 Biological Replicates (BR). 3 Technical replicates (TR1-TR3). Two biological replicates of time series with two technical replicates each forming a group of four wells in the last two. Each group has an order of magnitude increase in the number protein of copies/cell.

## Notes

1. If the salts are mixed and the pH is measured at this point it will be lower than 5.5. However, the Agar type can impact the final pH therefore care has to be taken to ensure a final pH of 5.5.

2. We have observed significant lower MBP-NL30F10H expression in BL21 Rosetta 2 pLyS at temperatures below 35 °C.

3. Pelleting the resuspension before freezing facilitates handling of samples because Rossetta 2 pLyS tends to burst and release genomic DNA which results in a significant viscosity increase. This can then be reduced by passing the crude lysate through a narrow-gauge blunt-ended syringe needle.

4. We have observed that storing 35S:NL3F10H at −80 °C results in a small decrease in activity (Urquiza-García and Millar, 2019). Therefore, we recommend maintaining the tissue in liquid nitrogen throughout the duration of the sampling and processing, or treating the calibration curve plants in exactly the same way (to control for changes in NanoLUC activity due to storage).

5. All photosynthetic organisms present a phenomenon called Delayed Fluorescence. This is luminescence that results after illumination due to processes related to the Photosystems II (PSII) and Cytochrome P680, which results in the emission of a photon. In seedlings, the signal is negligible after 60 s in darkness (Gould et al, 2009).

6. In order to facilitate interpretation of results, create a calibration curve that already represents the number of molecules per cell. It has been estimated that Col-0 fully expanded leaves contain on average 25 million cells per gram of fresh tissue (Flis et al, 2015). The user can create a standard curve with units of copies per cell by using this information plus the quantification of the NanoLUC standard, the molecular size of the NanoLUC standard, and the weight of the tissue collected.

7. Proteins related to the circadian clock and photobiology pathways can be highly unstable with high half-lives in the order of minutes. Therefore protease and proteasome inhibitors are absolutely required while processing the samples.

8. Some plates can become autofluorescent upon exposure to light. Therefore, in order to minimize background noise, use plates that have been designed for bioluminescent measurements (like Greiner Lumitrac plates). NanoLUC activity is so high that using white plates might result in contamination of neighbouring wells by high-emitting samples. Therefore, black plates provide a much better option given the strong signal emission of NanoLUC.

9. The NanoGlo® reagent has an emulsifier that is prone to generating foam. This might increase the experimental error during the assay.

## Data availability

The datasets, models and computer code produced in this study are available in the following resources: Data, models in multiple formats, construct maps, scripts, Docker container for the software environment and other resources, in a static archive, structured according to the standard ISA hierarchy, and formatted as a Research Object: FAIRDOMHub.org (https://doi.org/10.15490/FAIRDOMHUB.1.INVESTIGATION.570.1) and Zenodo repository (https://doi.org/10.5281/zenodo.14526989). The archive should be cited with its doi, as (for Zenodo): Urquiza-Garcia, U., Molina, N., Halliday K. J. and Millar, A. J. (2024). Absolute units in the circadian clock models of Arabidopsis up to U2020.5 [Data set]. Zenodo. https://doi.org/10.5281/zenodo.14526989. All the information above in a live, updatable resource: FAIRDOMHub.org (https://fairdomhub.org/investigations/570). Rhythmic expression data and period analysis used to select PRR7-complemented lines (Fig. EV1A–C; Table EV7): BioDare2 repository for chronobiology data (Zielinski et al, 2022); ID 10848; prr79 CCR2:LUC period complementation using PRR7NL constructs (https://biodare2.ed.ac.uk/experiment/10848); Rhythmic expression data and period analysis used to select CCA1-, LHY- and TOC1-complemented lines (Fig. EV1A–C; Table EV7): BioDare2 ID 11139; Complementation experiment lines CCA1NL, LHYNL, TOC1NL (https://biodare2.ed.ac.uk/experiment/11139); Rhythmic expression data and period analysis used to select LUX-complemented lines (Fig. EV1A–C; Table EV7): BioDare2 ID 11043; LUXp:LUX-NL3F10H; (https://biodare2.ed.ac.uk/experiment/11043); Rhythmic expression data of NanoLUC fusions in vivo under Light:Dark cycles (Fig. 6 and EV3): BioDare2 ID 11391; Plate reader experiment CCA1 TOC1 NanoLUC (https://biodare2.ed.ac.uk/experiment/11391). Step by step protocol for absolute quantification of proteins using NanoLUC fusion reporters in transgenic plants: Protocols.io (https://doi.org/10.17504/protocols.io.4r3l29n4jv1y/v1).

The source data of this paper are collected in the following database record: biostudies:S-SCDT-10_1038-S44320-025-00086-5.

## Peer review information

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

## Acknowledgements

For the purpose of open access, the author has applied a Creative Commons Attribution (CC BY) licence to any Author Accepted Manuscript version arising from this submission. For their expert assistance, we thank the plant growth facility, especially Sophie Haupt; Sarah Hodge with ultra-low-light imaging, Katalin Kis with laboratory methods and Dr. Marissa Valdivia Cabrera for curation of transgenic lines. For generously sharing materials, we thank Prof. Tsuyoshi Nakagawa (Shimane University, Japan) for providing pGWB vectors, Prof. Seth Davis (York University, UK) for the ABI strain and Prof. Norihito Nakamichi (Nagoya University, Japan) for sharing seeds of Col-0 CCA1-LUC lines. This work was funded in part by European Commission FP7 collaborative project TiMet (contract 245143) to AJM and others, and by United Kingdom Research and Innovation, Biotechnology, and Biological Sciences Research Council Grant BB/M025551/1 to KJH. UU-G was first funded by Ph.D. scholarship 216707 from the Consejo Nacional de Ciencia y Tecnología (CONACYT, México). UU-G was later funded by the Deutsche Forschungsgemeinschaft (DFG, German Research Foundation) under Germany's Excellence Strategy – EXC-2048/1 – project ID 390686111.

## Author contributions

**Uriel Urquiza-García**: Conceptualization; Resources; Data curation; Software; Formal analysis; Funding acquisition; Investigation; Visualization; Methodology; Writing—original draft; Writing—review and editing. **Nacho Molina**: Formal analysis; Methodology; Writing—review and editing. **Karen J Halliday**: Supervision; Funding acquisition; Writing—review and editing. **Andrew J Millar**: Conceptualization; Data curation; Supervision; Funding acquisition; Validation; Visualization; Writing—original draft; Project administration; Writing—review and editing.

Source data underlying figure panels in this paper may have individual authorship assigned. Where available, figure panel/source data authorship is listed in the following database record: biostudies:S-SCDT-10_1038-S44320-025-00086-5.

## Disclosure and competing interests statement

The authors declare that they have no conflict of interest. The authors' training and expertise comprise molecular genetics (UUG, KJH, AJM), genomics and mathematics (UUG, NM), biological physics (NM), plant physiology (KJH), systems and circadian biology (all authors) and research data management (UUG, AJM). The Centre for Engineering Biology (formerly SynthSys, and the Centre for Systems Biology at Edinburgh) is an interdisciplinary research centre. AJM was its founding Director. The University of Edinburgh is one of the largest research-intensive universities in the United Kingdom.

# Expanded View Figures

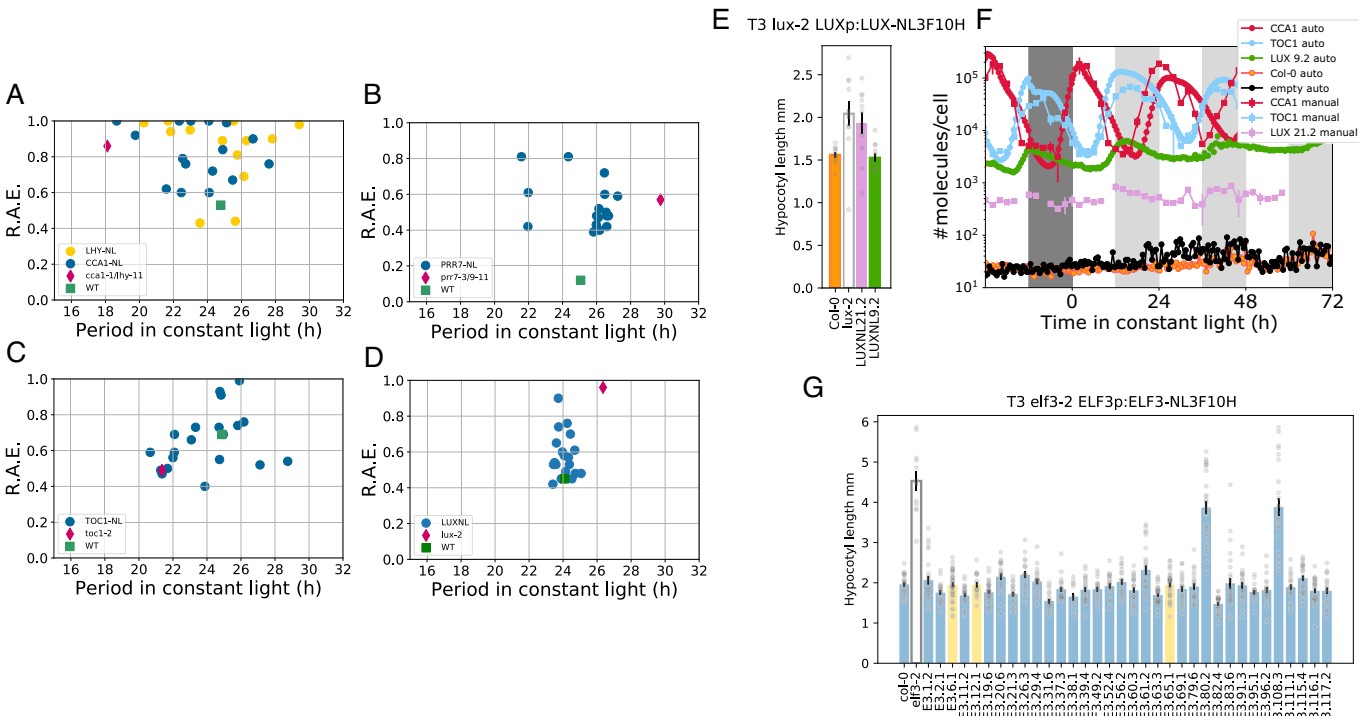

**Figure EV1.  Reporter fusion constructs rescue clock mutant phenotypes.**

Each NanoLUC protein reporter line was selected for the rescue of a clock gene mutant phenotype. (**A–D**) Circadian rhythms in reporter lines were monitored by in vivo imaging of seedlings under constant light, using the firefly LUC transcriptional reporter included in the background of each mutant. Each data point represents the period and relative amplitude error (R.A.E.) of a group of seedlings (*n* = 10) from an independent, single-insert, homozygous line in the T3 generation for the constructs listed, compared to the mutant host (red diamond) and wild type (green square) controls. (**A**) CCA1p:CCA1-NL3F10 (blue) and LHYp:LHY-NL3F10H (yellow) in the mutant background *cca1-1/lhy-11 CCA1p:LUC*. (**B**) PRR7p:PRR7-NL3F10H in *prr7-3/prr9-11 CCR2:LUC*. (**C**) TOC1p:TOC1-NL3F10H in *toc1-2 CCA1p:LUC*. (**D**) LUX2p:LUX-NL3F10H in *lux-2 CAB2:LUC*. (**E–G**) In a second round of selection, (**E**) a further LUX protein reporter line was selected (line 9.2, green), which was complemented to the Col-0 hypocotyl length (orange) whereas the line 21.2 (pink) retained the long hypocotyl phenotype of the *lux-2* mutant parent (open bar). (**F**) In vivo data ('auto') from line 9.2 under LD and LL (green), after the same detrending and rescaling as for CCA1 (red) and TOC1 (blue) (see Methods), suggested an expression level about 7-fold higher than line 21.2 (pink) tested in extracts ('manual'). Col-0 controls (orange) had the same low background signal as an empty well (black). (**G**) the ELF protein reporter lines (yellow) were selected based on the hypocotyl phenotype rescued to match Col-0 (far left, blue) and clearly rescue the long hypocotyl of the *elf3-2* mutant control (open bar).

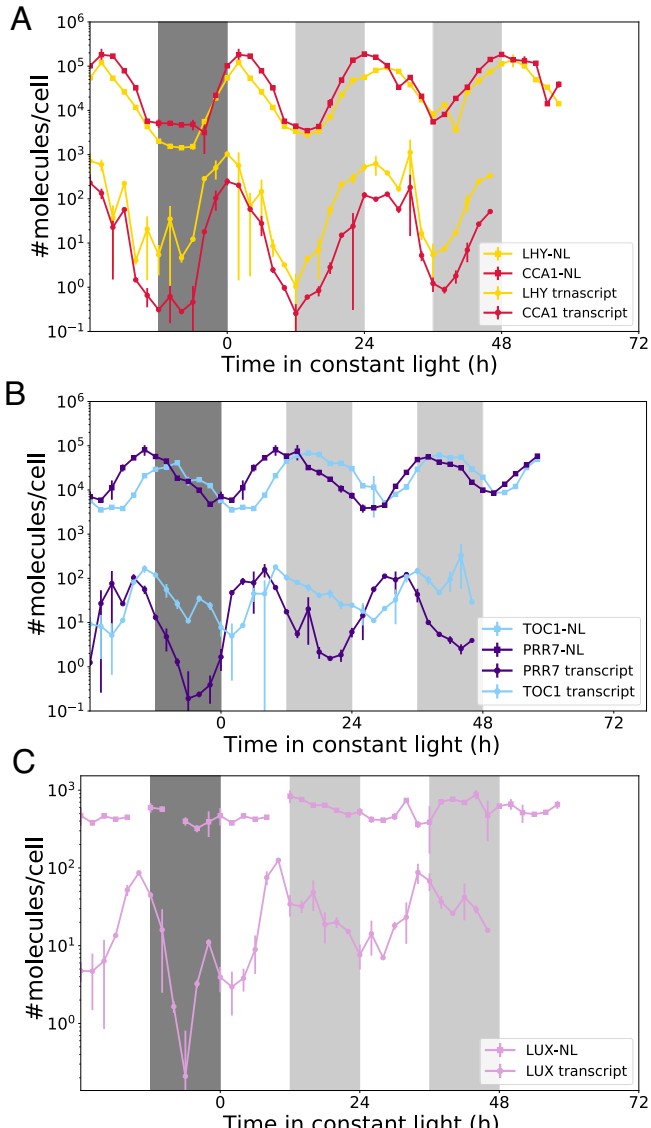

**Figure EV2.  Timeseries of clock protein copy numbers relative to mRNA.**

Reporter protein levels in plant extracts (data as in Fig. 6) were measured in calibrated NanoLUC assays for (**A**) LHY and CCA1, (**B**) PRR7 and TOC1 and (**C**) LUX line 21.2, under a 12L:12D cycle followed by constant light from time 0 h. Note that this LUX line was only partially complemented, see Appendix. Protein levels are compared to RNA levels in the TiMet data set, each in units of molecules per cell. Protein data are means of biological triplicates, RNA data are means of duplicates, error bar = 1 SEM. Light interval, white background; dark interval, dark grey shading; anticipated dark interval during constant light, light grey shading.

A

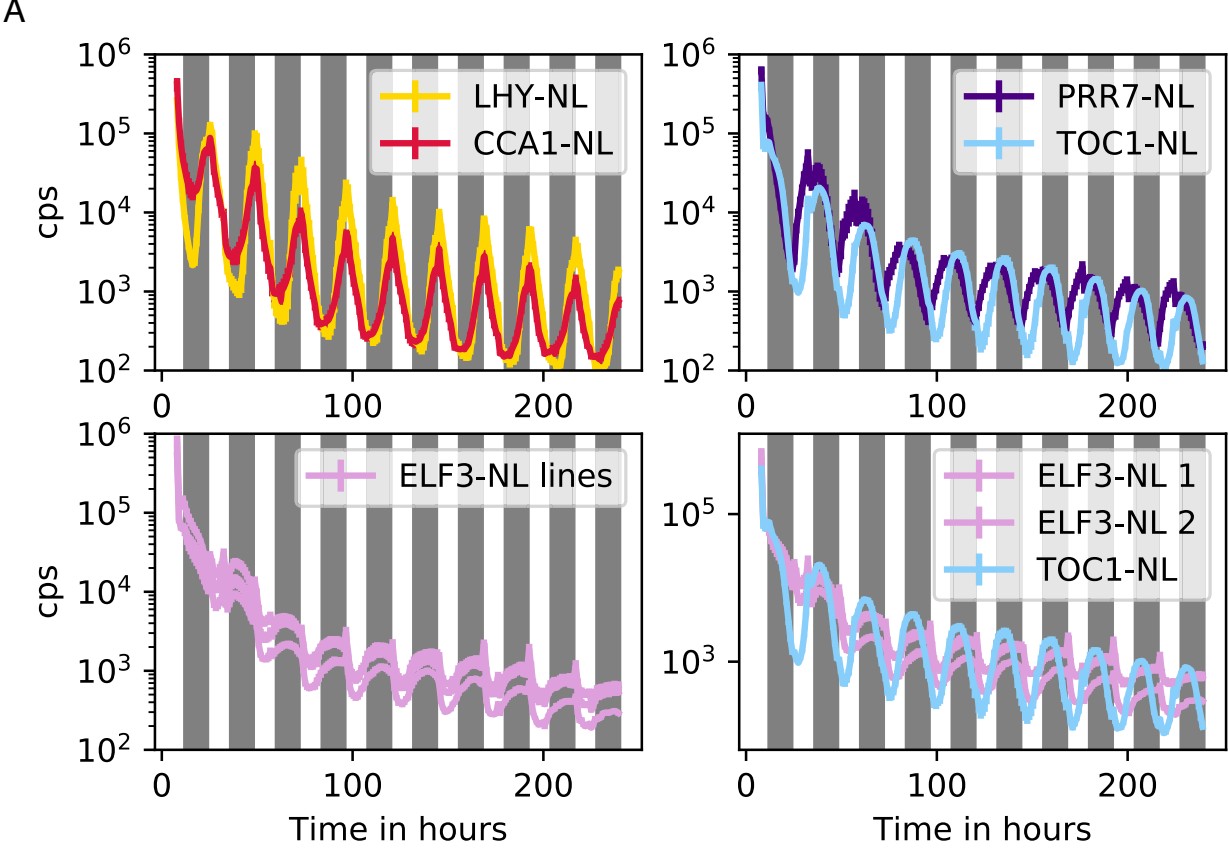

B

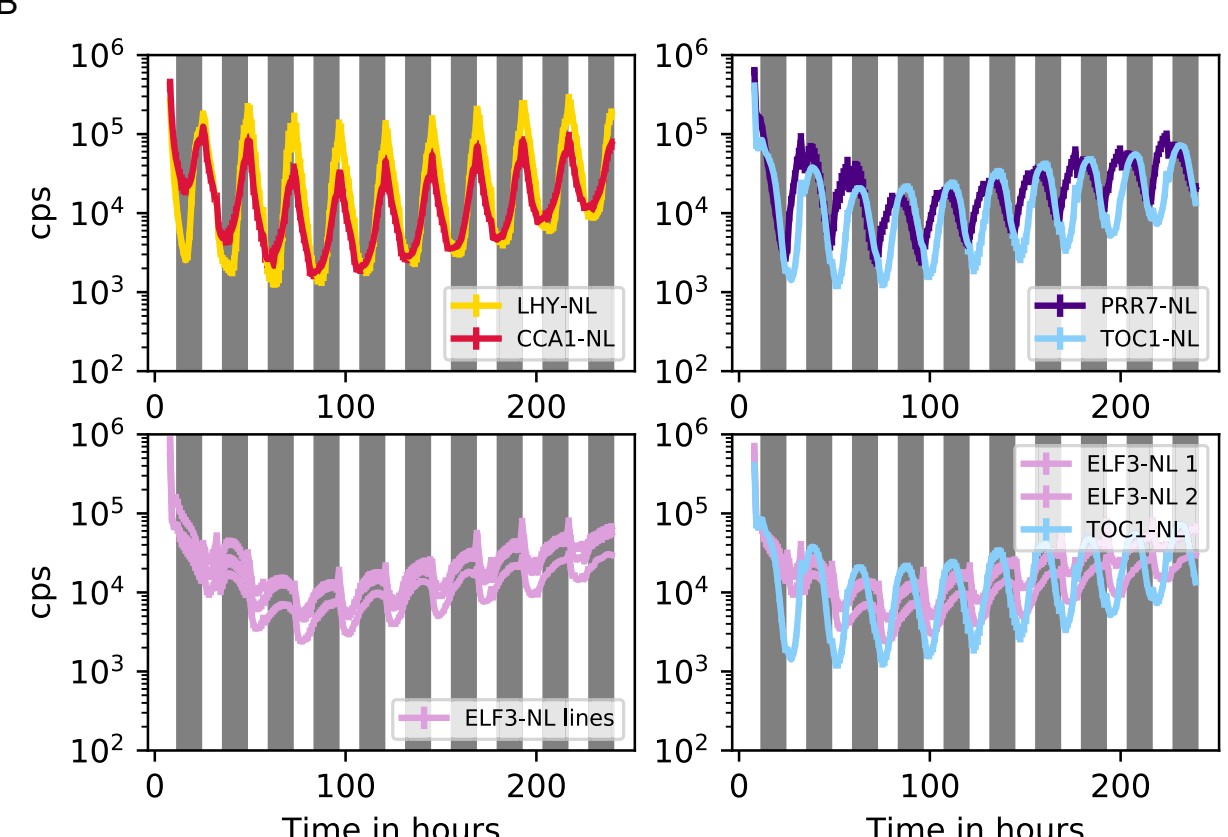

◄ **Figure EV3.  Long-term monitoring of protein fusions in vivo.**

(A) NanoLUC activity was measured (as in Fig. 6D, E) over ten 12 L:12D cycles, in seedlings carrying the protein reporters indicated, for LHY (yellow), CCA1 (red), PRR7 (purple), TOC1 (cyan) and in three ELF3 reporter lines (pink), hourly in an automated luminometer. Each trace is from a micro-well plate seeded with 4 seeds per well and incubated under ten, 12L:12D cycles, treated with furimazine substrate and assayed for a further 10 cycles. (B) The falling trend due to furimazine decay was removed from the in vivo signals. A single exponential decay rate was estimated from a large data set of the CCA1 protein reporter and applied to detrend all the datasets. All the lines show acute responses to light-dark transitions.

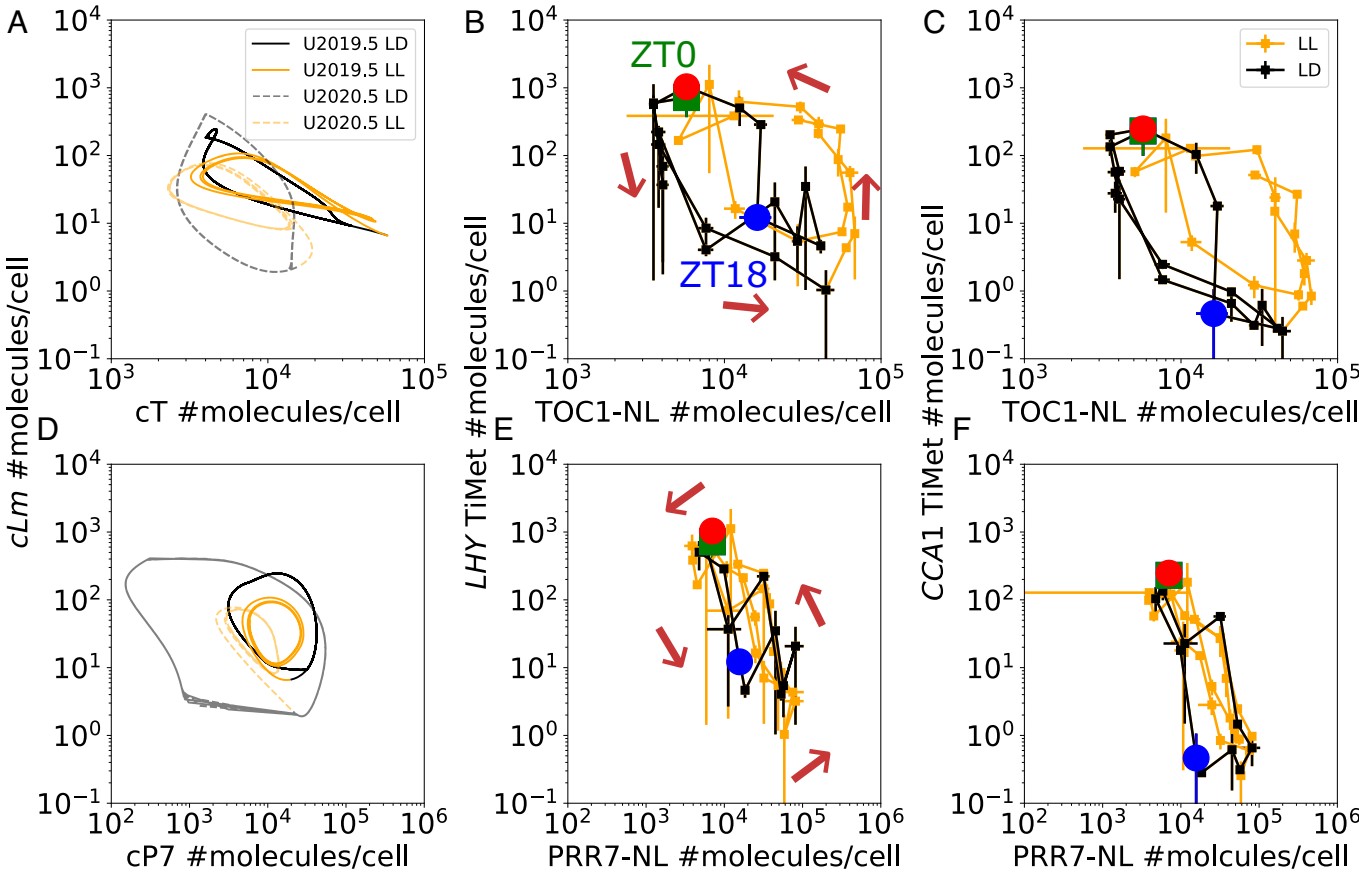

**Figure EV4. Regulation of *LHY* and *CCA1* by TOC1 and PRR7.**

Phase plane diagrams compare the accumulation of PRR transcriptional repressor proteins compared to their target mRNAs (as in Fig. 7). (**A**) Variables *cT* and *cLm* in models U2019.5 (dashed lines) and U2020.5 (solid lines), (**B**) TOC1 levels in extracts (as in Fig. 6) and TiMet *LHY* mRNA data, (**C**) TOC1 levels in extracts and TiMet *CCA1* mRNA data (Flis et al, 2015), under 12L:12D cycles (black lines) and constant light (yellow lines). TOC1 levels are lower when *CCA1* mRNA levels in 12L:12D than under LL (**C**), as for *LHY* mRNA levels (**B**). (**D**) Variables *cP7* and *cLm* in models U2019.5 (dashed lines) and U2020.5 (solid lines), (**E**) PRR7 levels in extracts (as in Fig. 6) and TiMet *LHY* mRNA data, (**F**) PRR7 levels in extracts and TiMet *CCA1* mRNA data, under 12L:12D cycles (black lines) and constant light (yellow lines). These variables are anti-correlated in the data (**E**, **F**) but plot a more circular cycle orbit in the model simulations (**D**). The simulated orbits also contract under LL (**D**), whereas both mRNA and protein amplitudes are maintained in the data (**E**, **F**). Markers in (**B**) show the first (green) and second (red) ZT0 (lights-on) and the intervening ZT18 (mid-night) under 12L:12D, and the direction of time (arrows). The last data point in black is ZT12 under 12L:12D. Error bars, 1 SEM.

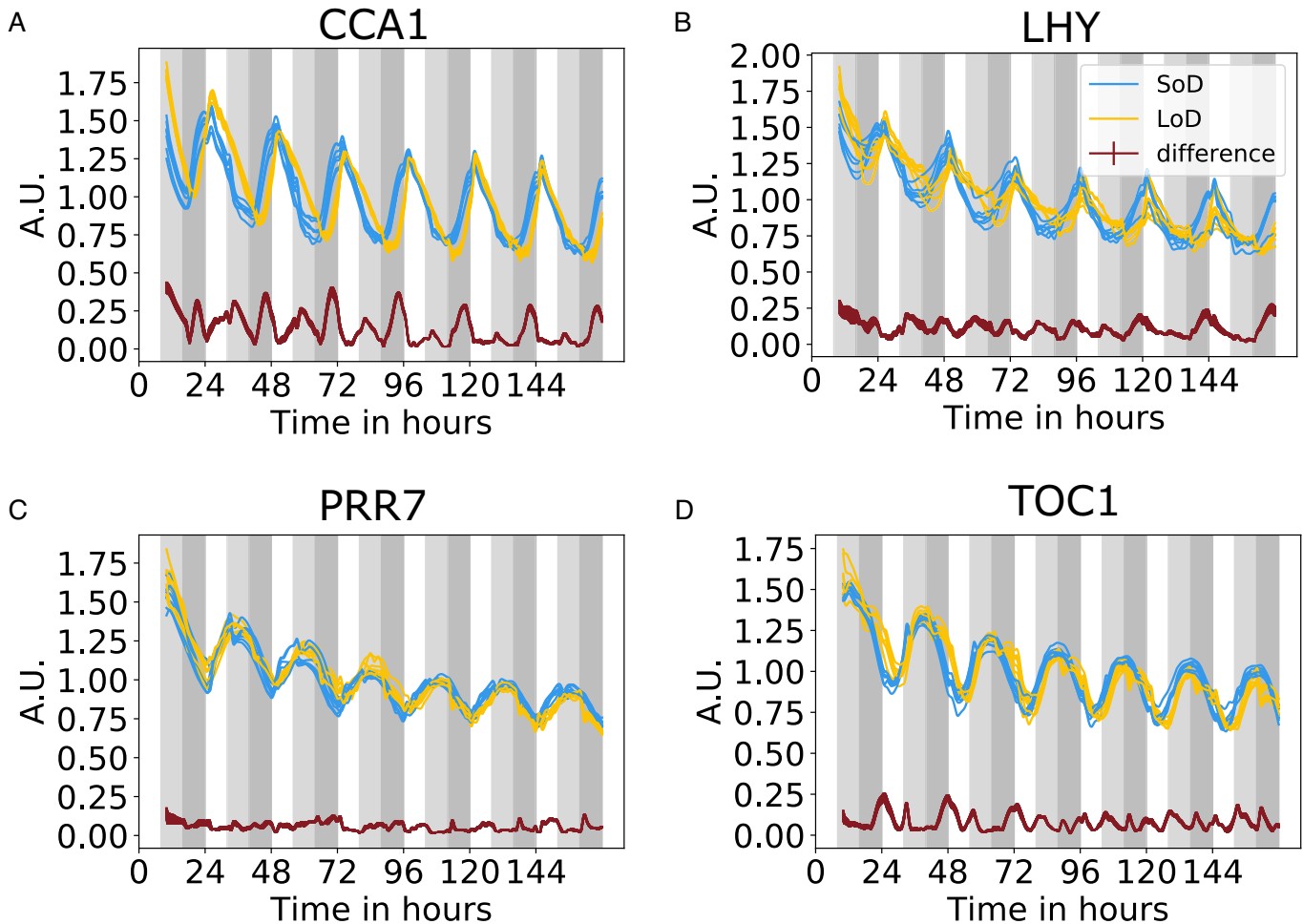

**Figure EV5.  Clock protein dynamics respond to photoperiod in vivo.**

In vivo recording reflects expected light-responsiveness, under short (8L:16D, cyan lines; SoD) compared to long photoperiods (16L:8D, yellow lines; LoD), as in Fig. 7C. Seedlings carrying reporter protein fusions to (**A**) CCA1, (**B**) LHY, (**C**) PRR7 and (**D**) TOC1 were grown for 10 of SoD or LoD in a multi-well plate and recorded hourly for 7 days in the same conditions, using an automated luminometer. Data were log-transformed and normalised to the mean of each timeseries (giving arbitrary units, A.U.). Data points for each well were connected with a cubic spline interpolation to facilitate comparison despite slight differences in sampling times. The absolute difference between the means (red line) shows the earlier rise of expression in the night under 8L:16D. White background, light interval; light grey, dark interval in 8L:16D only; dark grey, dark in both conditions.

