## [Peer Review File · Molecular Systems Biology]

Abundant clock proteins point to missing molecular regulation in the plant circadian clock

Uriel Urquiza-Garcia, Nacho Molina, Karen Halliday, and Andrew Millar

Corresponding author(s): Andrew Millar (Andrew.Millar@ed.ac.uk)

Review Timeline:	Transfer Date:	13th Nov 24
	Editorial Decision:	13th Dec 24
	Revision Received:	20th Dec 24
	Accepted:	3rd Jan 25

Editor: Poonam Bheda

Transaction Report: This manuscript was transferred to Molecular Systems Biology following peer review at Review Commons.

Review #1

1. Evidence, reproducibility and clarity:

Evidence, reproducibility and clarity (Required)

****Major****

1. In this work, the authors observed discrepancies between the estimated and actual values of DNA-binding dissociation constant (Kd). Such inconsistencies may arise if the model used to simulate Kd omits certain regulatory mechanisms. For example, previous studies (<https://doi.org/10.1371/journal.pcbi.1008340>, <https://doi.org/10.1098/rsfs.2021.0084>) showed that when specific gene regulatory mechanisms are missing, the dissociation constant required for generating oscillations can be underestimated. Thus, the inconsistency between the estimated and actual Kd values might result from missing mechanisms in the model. It would be beneficial if the authors could discuss this possibility.

****Minor****

1. The order in which the figures are mentioned does not match the order of the figures. Please adjust the sequence of the figures accordingly.
2. In the last paragraph of Introduction, the authors state, 'The absolute numbers of proteins directly constrain their possible biochemical activities (Kim & Forger, 2012)'. It would be helpful to also cite Jeong et al. (<https://doi.org/10.1073/pnas.2113403119>), who showed that two circadian neuronal groups in *Drosophila*, containing different amount of clock proteins, exhibit distinct molecular properties within the circadian clock.
3. In the title of the first section of Results, 'Predicting clock proteins levels' should be revised to 'Predicting clock protein levels.'
4. The description of the results in Figure 4 is unclear. Please, refer to the color and shape of the points when explaining the results for better clarity.

2. Significance:

Significance (Required)

In the manuscript by U. Urquiza-Garcia et al., entitled "Abundant clock proteins point to missing molecular regulation in the plant circadian clock," the authors refactored two models of the plant circadian clock to use absolute units, allowing direct comparison with biochemical data. This study significantly advances the integration of experimental data into computational models.

3. How much time do you estimate the authors will need to complete the suggested revisions:

Estimated time to Complete Revisions (Required)

(Decision Recommendation)

Less than 1 month

4. Review Commons values the work of reviewers and encourages them to get credit for their work. Select 'Yes' below to register your reviewing activity at Web of Science Reviewer Recognition Service (formerly Publons); note that the content of your review will not be visible on Web of Science.

Yes

Review #2

1. Evidence, reproducibility and clarity:

Evidence, reproducibility and clarity (Required)

1. ****Summary of this work****

This manuscript delves into the Arabidopsis circadian clock by addressing a major gap in plant gene circuit models-particularly, the lack of absolute units for protein quantification, which has historically hindered comparisons to biochemical data. By recalibrating two mathematical models of Arabidopsis thaliana's circadian clock from relative to absolute units, the authors introduce a framework that allows protein levels to be quantified in terms of copies per cell. This approach facilitates more direct comparisons between model predictions and empirical measurements.

The core of the study involves both analytical predictions and experimental validation to quantify protein levels for key clock proteins (such as LHY, CCA1, PRR7, and TOC1) using RNA data and luciferase reporter protein fusions. The simple model predicts the abundance of clock proteins, suggesting that the protein concentration may reach up to 100,000 copies per cell, which was then verified through experiments involving NanoLUC reporters. The recalibration of detailed mathematical models enabled the calculation of DNA-binding dissociation constants (Kd) based on empirical data, establishing a bridge between theoretical and experimental insights for understanding plant circadian rhythms.

The authors further explore how recalibrated models align with data by focusing on Kd values associated with DNA binding, particularly with LUX and CCA1 proteins. In vitro binding assays validated these predictions, while certain discrepancies emerged for CCA1 binding, implying other mechanisms could be influencing the observed results. Importantly, the recalibrated models provide a more realistic representation of protein-DNA interactions within the circadian clock, and the framework can be applied to understand other plant gene regulatory networks.

Overall, the authors demonstrate how absolute protein quantification can advance our understanding of circadian rhythm dynamics in Arabidopsis. This study highlights the broader applicability of their methods, suggesting potential adaptations for investigating gene regulation

across plant species and evolutionary contexts. The integration of empirical data with mathematical models introduces a new standard for rigor in plant gene circuit modeling and opens up avenues for exploring gene regulation in crop species and adaptive evolution in plants.

2. **Major points**

2.1. Limited Generalizability of Some Assumptions:

The translation and degradation rate assumptions are based on data from specific temperature and light conditions, which may not generalize well to other growth conditions. The authors could address this point by explicitly stating this limitation and explaining how variable conditions would affect the model's underlying assumptions.

2.2. Discrepancies in CCA1 Binding Affinity:

The model's simulated K_d values for CCA1 largely deviate from empirical measurements, suggesting missing mechanisms that may impact protein binding affinity. This might be due to some structural or environmental factors influencing CCA1 binding. No experiments are needed here but some plausible explanations would enhance the manuscript.

2.3. Limited Data for Evening Complex Proteins:

The model assumptions for ELF3, ELF4, and LUX rely on estimated degradation rates, which could introduce inaccuracies in the predictions. Empirical quantification of these rates would strengthen the reliability of these protein dynamics in the model. If these experiments are too resource and time intensive, then include some explanation of how changing these estimated degradation rates would affect the overall result.

2.4. Unexplained Variability in Reporter Data:

The NanoLUC reporter assays show some variability that the model does not account for, possibly due to factors like differential protein stability or folding in vivo. Further tests across different expression contexts, or including protein stability measurements, would clarify these inconsistencies. If these experiments are too resource and time intensive, then include some explanation of how changing these estimated degradation rates would affect the overall result.

2.5. Simplified Model for Translational Efficiency:

The model simplifies translational efficiency, which may not fully capture the complexity of clock protein synthesis. A more comprehensive approach, considering factors like ribosome density variation across transcripts, would add depth to the protein quantification model. Experiments are not required here. But it'd be nice to explain how a more complex model involving differential ribosomal density per transcript could affect the overall conclusions.

3. **Minor Points**

- Figure Legends: Several figures lack sufficient detail in legends, particularly Figures 3 and 5, where the methodology for generating the predicted protein levels and Kd values could be described a bit more, without majorly elongating the caption length.
- Unclear Units in Supplementary Table 1: Some units in Supplementary Table 1 for translation and degradation rates are not clearly specified.

2. Significance:

Significance (Required)

4. ****Overall Evaluation****

The recalibration of models to absolute units for protein quantification is a novel advancement in the field, allowing more direct comparison to experimental data. The study's combination of modeling and empirical validation is robust, and the use of quantitative NanoLUC reporters adds rigor to the experimental design. The study offers a clear protocol for estimating Kd values by integrating Protein-Binding Microarray (PBM) data and Surface Plasmon Resonance (SPR) data, which is a significant methodological contribution. These approaches could become a new standard for studies aiming to link molecular and phenotypic traits in plants. Moreover, through NanoLUC reporter assays, the study provides empirical data for protein levels that align closely with the model's predictions, enhancing the validity of their model. Additionally, by creating a framework that can be adapted to other plant gene regulatory networks, the authors extend the impact of their work beyond the Arabidopsis circadian clock, hinting at potential applications in agriculture and crop science.

I recommend this study with a minor revision, addressing the points below.

3. How much time do you estimate the authors will need to complete the suggested revisions:

Estimated time to Complete Revisions (Required)

(Decision Recommendation)

Less than 1 month

No

Review #3

1. Evidence, reproducibility and clarity:

Evidence, reproducibility and clarity (Required)

****Summary****

This is a valuable manuscript that begins to factorise the plant circadian model according to estimated protein translation and degradation rates. These models are iterative by nature, but the latest models provide a significant advance in our understanding and highlight avenues for future exploration. The use of Nanoluc protein:reporter fusions enables estimation of protein abundance, providing additional support for the modelled biological process.

The latest models highlight discrepancies between protein abundance predicted in the model compared to the experimental evidence, and sensible suggestions are discussed to prioritise experiments to enable greater biological understanding.

****Major comments****

1. I would have appreciated inclusion of an additional figure describing the U2019 and U2020 that were previously published by this group in 2021, along with a brief clarification of the differences between the simple and full versions of these models (Beyond the small summary presented in Figure 8).
2. It would have been useful to quantify the mRNA expression levels in the rescued transgenic lines to enable direct comparison to WT. The use of multiple independent transgenic lines supports the authors' conclusions but this characterisation would aid future research.

****Minor comments****

Supplemental figure 2 is partially cut off

It would be useful if figures/supplemental figures were provided in order.

2. Significance:

Significance (Required)

This work extends models of the plant circadian system to assess absolute protein numbers. This enables the biological 'plausibility' of the model to be assessed and also highlights where the model diverges from experimental data, indicating where additional understanding is required.

3. How much time do you estimate the authors will need to complete the suggested revisions:

Estimated time to Complete Revisions (Required)

(Decision Recommendation)

Less than 1 month

Yes

Revision Plan

Manuscript number: RC-2024-02679

Corresponding author(s): Andrew MILLAR

1. General Statements

We are very grateful for the three reviewers' positive and considered comments. We **copy specific comments below in bold font**, and address them in turn including *quotations from the revised manuscript in italics*.

2. Description of the planned revisions

One Supplementary Figure could be copied from an earlier manuscript to show the model circuits, please see Reviewer 3 comment 1, below.

3. Description of the revisions that have already been incorporated in the transferred manuscript

Reviewer 1

Major

1. In this work, the authors observed discrepancies between the estimated and actual values of DNA-binding dissociation constant (Kd). Such inconsistencies may arise if the model used to simulate Kd omits certain regulatory mechanisms. For example, previous studies (<https://doi.org/10.1371/journal.pcbi.1008340>, <https://doi.org/10.1098/rsfs.2021.0084>) showed that when specific gene regulatory mechanisms are missing, the dissociation constant required for generating oscillations can be underestimated. Thus, the inconsistency between the estimated and actual Kd values might result from missing mechanisms in the model. It would be beneficial if the authors could discuss this possibility.

We thank the reviewer for these examples, which illustrate the relevant point that multiple biochemical processes can contribute to the non-linearity required to simulate a particular clock behaviour within biochemically-reasonable parameter values. We now cite both, along with the original Buchler & Louis paper on the titration mechanism in general, as follows:

"It is possible that the measured, bulk clock protein levels might over-estimate the protein available for promoter binding due to mechanisms absent from the model, such as protein partitioning outside the nucleus (Yakir et al, 2009), protein titration (Buchler & Louis, 2008), clustering of proteins within the nucleus, a processing step akin to the formation of a smaller EC pool from a fraction of the bulk LUX protein or any combination of these mechanisms (Jeong et al, 2022b; Yao et al, 2022)."

Minor

1. The order in which the figures are mentioned does not match the order of the figures. Please adjust the sequence of the figures accordingly.

- Done. The amalgamated PDF for review presented the Figures and Supplementary Figures in the order of citation, which we intended for the reviewers' convenience. As this didn't help two of the reviewers, they are now in numerical sequence.

2. In the last paragraph of Introduction, the authors state, 'The absolute numbers of proteins directly constrain their possible biochemical activities (Kim & Forger, 2012)'. It would be helpful to also cite Jeong et al. (<https://doi.org/10.1073/pnas.2113403119>), who showed that two circadian neuronal groups in *Drosophila*, containing different amount of clock proteins, exhibit distinct molecular properties within the circadian clock.

- Added a citation to this relevant example later in that sentence.

3. In the title of the first section of Results, 'Predicting clock proteins levels' should be revised to 'Predicting clock protein levels.'

- Corrected.

4. The description of the results in Figure 4 is unclear. Please, refer to the color and shape of the points when explaining the results for better clarity.

- The legend of Figure 4 now refers to both characteristics.

Reviewer 2

2.1. Limited Generalizability of Some Assumptions:

The translation and degradation rate assumptions are based on data from specific temperature and light conditions, which may not generalize well to other growth conditions. The authors could address this point by explicitly stating this limitation and explaining how variable conditions would affect the model's underlying assumptions.

- To address this point, we now note in Discussion: *"For example, the reporter assays could quickly test protein numbers under different conditions from those reported here, to understand the biochemical mechanisms for the canonical 'temperature compensation' of circadian period in constant conditions (as modelled in Gould et al, 2013) and/or the adaptation to fluctuating, natural conditions (see Future perspectives: models informed by genome sequence, below)."*

2.5. Simplified Model for Translational Efficiency:

The model simplifies translational efficiency, which may not fully capture the complexity of clock protein synthesis. A more comprehensive approach, considering factors like ribosome density variation across transcripts, would add depth to the protein quantification model. Experiments are not required here. But it'd be nice to explain how a more complex model involving differential ribosomal density per transcript could affect the overall conclusions.

- The 'simple model' is intentionally simple. We note in the Results that it even ignores the published light/dark-regulation of translation rate, both generally in Arabidopsis and specifically for LHY.

To address this point, we have added in Discussion, *"In other words, the bulk levels of these clock proteins might be rather simply regulated. The simple model's approach could justifiably be repeated to estimate the levels of other proteins, and extended to test where more complex biochemical mechanisms, such as translational regulation, are functionally significant."*

3. Minor Points

- Figure Legends: Several figures lack sufficient detail in legends, particularly Figures 3 and 5, where the methodology for generating the predicted protein levels and Kd values could be described a bit more, without majorly elongating the caption length.

- Both legends have been updated. However, the methods are described in detail in the Supplementary Information, which provides much more space.

- Unclear Units in Supplementary Table 1: Some units in Supplementary Table 1 for translation and degradation rates are not clearly specified.

- Units seem clear in the column headings, as follows:

s (proteins per mRNA per hour)	k or k light (per hour)	k dark (per hour)
--	---------------------------------

The reviewer might be highlighting that some values of *k**dark* were given as "-" because that protein did not have light-regulated degradation in the simple model, so degradation rate was just *k* rather than *k**light* and *k**dark*. This notation has been updated to NA, with an explanatory note in the supplementary table legend.

Reviewer 3

1. I would have appreciated inclusion of an additional figure describing the U2019 and U2020 that were previously published by this group in 2021, along with a brief clarification of the differences between the simple and full versions of these models (Beyond the small summary presented in Figure 8).

Supplementary Figure 3 of Urquiza and Millar 2021 shows simplified circuit diagrams of the two models and could be added as a new Supplementary Figure in this manuscript, as the editor prefers. To address this point, we added an introduction to the detailed models in the Results: *"Our detailed models of the clock gene circuit (see Introduction) are not driven by rhythmic data input like the simple model, but rather use ordinary differential equations to recapitulate the dynamics of each RNA and protein component in the clock circuit, along with their interconnected feedback loops and their regulation by light signals. The models autonomously generate rhythmic patterns of RNA and protein expression that match the rhythmic data. The gene circuits in models U2019 and U2020 differ only in the regulation of daytime processes, involving LHY/CCA1 and the PRR genes (Urquiza-García & Millar, 2021). The circuit of U2019 is closer to its antecedent model P2011 (Pokhilko et al, 2011), using gene activation, whereas*

U2020 uses repression (for circuit diagrams, see Supplementary Figure 3 of (Urquiza-García & Millar, 2021). Repression is better supported by molecular data but U2020 simulations fit the data no better than, or slightly worse than, the activation-based model U2019 (consistent with Fogelmark & Troein, 2014), so we use both circuits here.”

Minor comments

Supplemental figure 2 is partially cut off

- Corrected. Only the edge of a label was affected.

It would be useful if figures/supplemental figures were provided in order.

- Corrected, see reviewer 1 above.

4. Description of analyses that authors prefer not to carry out

Reviewer 1

N/A

Reviewer 2

2.2. Discrepancies in CCA1 Binding Affinity:

The model's simulated K_d values for CCA1 largely deviate from empirical measurements, suggesting missing mechanisms that may impact protein binding affinity. This might be due to some structural or environmental factors influencing CCA1 binding. No experiments are needed here but some plausible explanations would enhance the manuscript.

- This point is addressed in the section 'Future perspectives: investigating K_d *in vivo*', now enhanced by our response to reviewer 1's major comment, as noted above.

2.3. Limited Data for Evening Complex Proteins:

The model assumptions for ELF3, ELF4, and LUX rely on estimated degradation rates, which could introduce inaccuracies in the predictions. Empirical quantification of these rates would strengthen the reliability of these protein dynamics in the model. If these experiments are too resource and time intensive, then include some explanation of how changing these estimated degradation rates would affect the overall result.

- We assume that the reviewer is referring to the k parameters of the simple model driven by RNA data, as in Figure 2, which gives the predicted protein levels that we compare to measured protein levels in Table 1. Both the prior degradation rate data and our NanoLUC reporter measurements are strongest for LHY, CCA1, PRR7 and TOC1, so these are our focus. New measurements of degradation rate for ELF3, ELF4 and LUX are indeed beyond our scope. Our reporter methods might facilitate future measurement of such parameters by other researchers.

In the Supplementary Information, we outline the estimation of degradation rates k , which returned biologically plausible protein half-lives for LUX = 3.7 h, ELF4 = 1.3 h and

ELF3 8.7 h. We have no data for ELF4 and we highlight the specific limitations of our ELF3 and LUX data in the Results and Discussion text, Table 1, Methods and Supplementary Information. Taking the *in vivo* data for ELF3 for example (Table 1), the protein levels predicted by the simple model using that degradation rate were within 6% to 7% of the measured levels at the peak and at the trough under LD. This independent result supports the estimated degradation rate: varying the estimated degradation rate for ELF3 would not maintain a prediction so remarkably close to the data.

2.4. Unexplained Variability in Reporter Data:

The NanoLUC reporter assays show some variability that the model does not account for, possibly due to factors like differential protein stability or folding in vivo. Further tests across different expression contexts, or including protein stability measurements, would clarify these inconsistencies. If these experiments are too resource and time intensive, then include some explanation of how changing these estimated degradation rates would affect the overall result.

- This comment cannot be addressed without knowing which of the many possible data series the reviewer is referring to, and whether the variability of interest is in level, timing or a higher-order dynamic behaviour. For example, we discuss three specific instances where reporter and model behaviour differs, in detail, in Discussion section 'Refining the modelled protein profiles' (726 words).

Reviewer 3

2. It would have been useful to quantify the mRNA expression levels in the rescued transgenic lines to enable direct comparison to WT. The use of multiple independent transgenic lines supports the authors' conclusions but this characterisation would aid future research.

The proposed study most directly addresses whether RNA from the reporter transgene is functionally equivalent to wild-type RNA, whereas the focus of this article is at the protein level. As the reviewer notes, the proposed study would principally be an aid to future research, so we prefer not to start that additional experimental work.

13th Dec 2024

Manuscript Number: MSB-2024-12754-T

Title: Abundant clock proteins point to missing molecular regulation in the plant circadian clock

Dear Dr. Millar,

Thank you for the submission of your revised manuscript to Molecular Systems Biology. We have now received the enclosed reports from the referees that were asked to re-assess it. As you will see the reviewers are now globally supportive and I am pleased to inform you that we will be able to accept your manuscript pending the following final amendments:

- 1) Please download the EMBO Press "Author Checklist" and complete all relevant questions. This file should be uploaded with your submission. This file can be downloaded from our website at:
<https://www.embopress.org/page/journal/17444292/authorguide>
- 2) Please upload a .docx version of the manuscript with no track changes.
- 3) In the main manuscript file, please include keywords to max. 5.
- 4) "Character count", "List of Supplementary Tables" and "Supplementary Information, contents" should be removed from the manuscript file.
- 5) Please ensure that the models in FAIRDOMHub are freely publicly available, as currently they are not. Please also be sure to include a README file with practical use instructions for potential future users of your code/model.
- 6) Please format the Data availability section describing how the data, code that were specifically generated for this study have been made available. Datasets that are reused from other studies should be listed in the Methods and potentially referenced as Data Citations (see below). Please format the Data Availability section according to the example below:
"The datasets and computer code produced in this study are available in the following databases:
- Chip-Seq data: Gene Expression Omnibus GSE46748 (<https://www.ncbi.nlm.nih.gov/geo/query/acc.cgi?acc=GSE46748>)
- Modeling computer scripts: GitHub (<https://github.com/SysBioChalmers/GECKO/releases/tag/v1.0>)
- [data type]: [full name of the resource] [accession number/identifier] ([doi or URL or identifiers.org/DATABASE:ACCESSION])"
- 7) Our journal encourages inclusion of *data citations in the reference list* to directly cite datasets that were re-used and obtained from public databases. Data citations in the article text are distinct from normal bibliographical citations and should directly link to the database records from which the data can be accessed. In the main text, data citations are formatted as follows: "Data ref: Smith et al, 2001" or "Data ref: NCBI Sequence Read Archive PRJNA342805, 2017". In the Reference list, data citations must be labeled with "[DATASET]". A data reference must provide the database name, accession number/identifiers and a resolvable link to the landing page from which the data can be accessed at the end of the reference. Further instructions are available at .
- 8) Please rename "Conflict of Interests" to "Disclosure and competing interests statement". We updated our journal's competing interests policy in January 2022 and request authors to consider both actual and perceived competing interests. Please review the policy <https://www.embopress.org/competing-interests> and update your competing interests if necessary.
- 9) Author contributions: Please remove it from the manuscript and specify author contributions in our submission system. CRedit has replaced the traditional author contributions section because it offers a systematic machine-readable author contributions format that allows for more effective research assessment. You are encouraged to use the free text boxes beneath each contributing author's name to add specific details on the author's contribution. More information is available in our guide to authors:
<https://www.embopress.org/page/journal/17574684/authorguide#authorshipguidelines>
<https://www.embopress.org/page/journal/17574684/authorguide#referencesformat>
- 10) In the Methods, please take care of the following:
 - The Materials and Methods section should be renamed to "Methods".
 - Please ensure that a statement on whether or not blinding was done is included in the Methods even if no blinding was done. Please also be sure to update the Author Checklist with this information and where it can be found in the manuscript.
- 11) All Materials and Methods need to be described in the main text using our 'Structured Methods' format. According to this format, the Methods section includes a Reagents and Tools Table (listing key reagents, experimental models, software and relevant equipment and including their sources and relevant identifiers) followed by a Methods and Protocols section describing the methods, ideally using a step-by-step protocol format. The aim is to facilitate adoption of the methodologies across labs. Please download and fill our Reagents and Tools Table template (.docx), which you can find in our author guidelines:
<https://www.embopress.org/page/journal/14693178/authorguide#structuredmethods>.
When submitting your revised manuscript, please do not include the Reagents and Tools Table in the Methods section of the manuscript but upload it as a separate file choosing the file type "Reagent Table".
An example of a Method paper with Structured Methods can be found here:
<https://www.embopress.org/doi/10.15252/msb.20178071>. "
- 12) Please place individual sections of the manuscript in the following order: Title page - Abstract & Keywords - Introduction - Results - Discussion - Methods - Data Availability - Acknowledgements - Disclosure and Competing Interests Statement - References - Figure Legends - Expanded View Figure Legends.
- 13) For the figures and figure legends, please take care of the following:

- Fig. 2 has only A-D panels, but there are callouts for 2C-F
 - A callout is missing for Fig. 6D
 - all callouts should be listed sequentially
 - figures should be uploaded as individual, high-resolution Figure files; up to 5 supplementary figures can be made into Expanded View figures - in that case, please upload these as Figure EV1-EV5 with figure legends included below main figure legends in the manuscript file with the heading Expanded View Figures Legends; The other remaining supplementary figures should be compiled into an Appendix PDF with the nomenclature Appendix Figure Sx
 - Main figures and EV figures should be uploaded as individual, high-resolution files. Please check "Author Guidelines" for more information: <https://www.embopress.org/page/journal/17574684/authorguide#figureformat>
 - Please note that information related to n is missing in the legends of figures 1A, 6D-F; 7B.
 - Please note that the error bars are not defined in the legends of figures 1A, 6D-F.
- 14) Supplementary tables should be uploaded as individual tables renamed to Table EVx with legends uploaded above the table in each Excel file; only Supplementary Table 5 should be uploaded as Dataset EV1 with the legend uploaded as a separate sheet/tab in the Excel file as this is the most complex table. Please also be sure to update their callouts in main manuscript text.
- 15) The Appendix file needs to be in PDF format; the title page should contain "Appendix for + ms title" and a table of contents with the page numbers for the listed items; the nomenclature should be Appendix Figure Sx and Appendix Table Sx. Please also be sure to update their callouts in main manuscript text.
- 16) Funding: Please note that funding information should be given in the "Acknowledgements" section (not in its own separate section).
- 17) Synopsis:
- Synopsis image: Please provide a graphic that summarises the main findings of the manuscript on a glance and upload it as a high-resolution jpeg file 550 pixels wide x (300-600) pixels high.
 - Synopsis text: Please provide a short standfirst (maximum of 300 characters, including space), limit the bullet points to max. 5 and upload it as a separate .doc file. Please write the bullet points to summarise the key NEW findings. They should be designed to be complementary to the abstract - i.e. not repeat the same text. We encourage inclusion of key acronyms and quantitative information (maximum of 30 words / bullet point). Please use the passive voice.
 - Please check your synopsis text and image before submission with your revised manuscript. Please be aware that in the proof stage minor corrections only are allowed (e.g., typos).
- 18) Source Data: Please ensure that a completed Source Data checklist is uploaded as a Related Manuscript File (you will be contacted by my colleague Hannah Sonntag with further instructions on Source Data and the checklist that needs to be filled out). Source Data should be organized as a single source data file (zipped) per figure for main figures (all EV and/or Appendix figure Source Data can be included in a single folder), with the panels clearly visible in the folder structure instead of a single excel file for all Source Data. e.g. all the Source data files for figure 1 need to be saved in a single folder and this needs to be zipped and then uploaded as "SD figure 1.zip" file.
- 19) As part of the EMBO Publications transparent editorial process initiative (see our policy here: https://www.embopress.org/transparent-process#Review_Process), Molecular Systems Biology will publish online a Peer Review File (PRF) to accompany accepted manuscripts. This file will be published in conjunction with your paper and will include the anonymous referee reports, your point-by-point response and all pertinent correspondence relating to the manuscript. Let us know whether you agree with the publication of the PRF and as here, if you want to remove or not any figures from it prior to publication. Please note that the Authors checklist will be published at the end of the PRF.
- 20) Please provide a point-by-point letter INCLUDING my comments and your detailed responses (as Word file).

I look forward to reading a new revised version of your manuscript as soon as possible.

Yours sincerely,

Poonam Bheda, PhD
Scientific Editor
Molecular Systems Biology

Reviewer #1:

The authors have done a good job in addressing most of my comments. I support the publication of the revised manuscript in MSB.

Reviewer #2:

The authors have addressed my comments appropriately.

Reviewer #3:

The authors have made commendable efforts to address the comments and suggestions provided in the initial review. They have adequately addressed the major concerns, particularly by thoughtfully discussing the possibility that discrepancies between the estimated and actual DNA-binding dissociation constant values may stem from missing regulatory mechanisms in the model. Additionally, the order of the figures has been corrected, and typographical issue has been addressed. The manuscript is now well-organized and I am pleased to recommend its publication in Molecular Systems Biology.

Rev_Com_number: RC-2024-02679

New_manu_number: MSB-2024-12754-T

Corr_author: Millar

Title: Abundant clock proteins point to missing molecular regulation in the plant circadian clock

Dear Editors,

Thank you for your attention to the formatting of our manuscript from Review Commons. The uploaded version now has the following amendments (to your numbered points):

1) Please download the EMBO Press "Author Checklist" and complete all relevant questions. This file should be uploaded with your submission. This file can be downloaded from our website at:

<https://www.embopress.org/page/journal/17444292/authorguide>

DONE

2) Please upload a .docx version of the manuscript with no track changes.

DONE

3) In the main manuscript file, please include keywords to max. 5.

DONE

4) "Character count", "List of Supplementary Tables" and "Supplementary Information, contents" should be removed from the manuscript file.

DONE.

5) Please ensure that the models in FAIRDOMHub are freely publicly available, as currently they are not. Please also be sure to include a README file with practical use instructions for potential future users of your code/model.

DONE. The DOIs for both FAIRDOMHub and Zenodo snapshots are included in the article under Data Availability. The models are in SBML format (and in Antimony too), and can be simulated in many software packages of the user's choosing. There is no model code to describe.

6) Please format the Data availability section describing how the data, code that were specifically generated for this study have been made available. Datasets that are reused from other studies should be listed in the Methods and potentially referenced as Data Citations (see below). Please format the Data Availability section according to the example below:

"The datasets and computer code produced in this study are available in the following databases:

- Chip-Seq data: Gene Expression Omnibus GSE46748

(<https://www.ncbi.nlm.nih.gov/geo/query/acc.cgi?acc=GSE46748>)

- Modeling computer scripts: GitHub (<https://github.com/SysBioChalmers/GECKO/releases/tag/v1.0>)

- [data type]: [full name of the resource] [accession number/identifier] ([doi or URL or identifiers.org/DATABASE:ACCESSION])"

DONE.

7) Our journal encourages inclusion of *data citations in the reference list* to directly cite datasets that were re-used and obtained from public databases. Data citations in the article text are distinct from normal bibliographical citations and should directly link to the database records from which the data can be accessed. In the main text, data citations are formatted as follows: "Data ref: Smith et al, 2001" or "Data ref: NCBI Sequence Read Archive PRJNA342805, 2017". In the Reference list, data citations must be labeled with "[DATASET]". A data reference must provide the database name, accession number/identifiers and a resolvable link to the landing page from which the data can be accessed at the end of the reference. Further instructions are available at

<https://www.embopress.org/page/journal/17574684/authorguide#referencesformat>.

DONE. As this format does not apply to data provided in Supplementary Information, only one resource is relevant in our case, namely the 1001 genomes project, now added both as citation and DATASET reference.

8) Please rename "Conflict of Interests" to "Disclosure and competing interests statement". We updated our journal's competing interests policy in January 2022 and request authors to consider both actual and perceived competing interests. Please review the policy <https://www.embopress.org/competing-interests> and update your competing interests if necessary.

DONE

9) Author contributions: Please remove it from the manuscript and specify author contributions in our submission system. CRediT has replaced the traditional author contributions section because it offers a systematic machine-readable author contributions format that allows for more effective research assessment. You are encouraged to use the free text boxes beneath each contributing author's name to add specific details on the author's contribution. More information is available in our guide to authors: <https://www.embopress.org/page/journal/17574684/authorguide#authorshipguidelines> <https://www.embopress.org/page/journal/17574684/authorguide#referencesformat>

DONE.

UUG – Conceptualization, Data curation, Formal analysis, Funding acquisition, Investigation, Methodology, Software, Resources, Visualization, Writing – original draft, Writing – review & editing

NM - Formal analysis, Methodology, Writing – review & editing

KJH - Funding acquisition, Supervision, Writing – review & editing

AJM – Conceptualization, Data curation, Funding acquisition, Project administration, Supervision, Validation, Visualization, Writing – original draft, Writing – review & editing

10) In the Methods, please take care of the following:

- The Materials and Methods section should be renamed to "Methods".

DONE.

- Please ensure that a statement on whether or not blinding was done is included in the Methods even if no blinding was done. Please also be sure to update the Author Checklist with this information and where it can be found in the manuscript.

DONE, at the end of experimental Methods.

11) All Materials and Methods need to be described in the main text using our 'Structured Methods' format. According to this format, the Methods section includes a Reagents and Tools Table (listing key reagents, experimental models, software and relevant equipment and including their sources and relevant identifiers) followed by a Methods and Protocols section describing the methods, ideally using a step-by-step protocol format. The aim is to facilitate adoption of the methodologies across labs.

DONE. The method for calibrated NanoLUC quantification is provided as step by step protocol. This is the key data-generating method for the article, and the only one that has not been previously described in detail.

Please download and fill our Reagents and Tools Table template (.docx), which you can find in our author guidelines:

<https://www.embopress.org/doi/10.15252/msb.20178071>. “

DONE, file uploaded.

12) Please place individual sections of the manuscript in the following order: Title page - Abstract & Keywords - Introduction - Results - Discussion - Methods - Data Availability - Acknowledgements - Disclosure and Competing Interests Statement - References - Figure Legends - Expanded View Figure Legends.

DONE

13) For the figures and figure legends, please take care of the following:

- Fig. 2 has only A-D panels, but there are callouts for 2C-F

DONE

- A callout is missing for Fig. 6D

DONE.

- all callouts should be listed sequentially

DONE – with the exception that Figures 7A,7B must be discussed before the live imaging in 6D-6F is introduced.

- figures should be uploaded as individual, high-resolution Figure files; up to 5 supplementary figures can be made into Expanded View figures - in that case, please upload these as Figure EV1-EV5 with figure legends included below main figure legends in the manuscript file with the heading Expanded View Figures Legends; The other remaining supplementary figures should be compiled into an Appendix PDF with the nomenclature Appendix Figure Sx

DONE, as follows:

Old name	New name	
Supplementary Figure 1-3	Appendix Figure S1-S3	Callouts done in text, added to Appendix PDF
Supplementary Figure 4-8	Figure EV1-EV5	DONE in text, legend and filenames
Supplementary Table 1-4	Table EV1-EV4	DONE
Supplementary Table 5	Dataset EV1	DONE
Supplementary Table 6-8	Table EV5-EV7	DONE

- Main figures and EV figures should be uploaded as individual, high-resolution files. Please check "Author Guidelines" for more information:

<https://www.embopress.org/page/journal/17574684/authorguide#figureformat>

DONE.

- Please note that information related to n is missing in the legends of figures 1A, 6D-F; 7B.
- Please note that the error bars are not defined in the legends of figures 1A, 6D-F.

Figure 1 has no data. Figure 6D-F, DONE. Figure 7B, legend now clarifies that data are replotted from Figure 6C.

14) Supplementary tables should be uploaded as individual tables renamed to Table EVx with legends uploaded above the table in each Excel file; only Supplementary Table 5 should be uploaded as Dataset EV1 with the legend uploaded as a separate sheet/tab in the Excel file as this is the most complex table. Please also be sure to update their callouts in main manuscript text.

DONE. Each Table has a title in cell A2. Tables EV1 and EV4 have legends in cells A3. Please note, several tables use data alongside the main columns. New Tables EV2 and EV3 have these data in columns F and G, new Table EV4 has a set of data with references in columns M-P.

15) The Appendix file needs to be in PDF format; the title page should contain "Appendix for + ms title" and a table of contents with the page numbers for the listed items; the nomenclature should be Appendix Figure Sx and Appendix Table Sx. Please also be sure to update their callouts in main manuscript text.

DONE.

16) Funding: Please note that funding information should be given in the "Acknowledgements" section (not in its own separate section).

DONE.

17) Synopsis:

- Synopsis image: Please provide a graphic that summarises the main findings of the manuscript on a glance and upload it as a high-resolution jpeg file 550 pixels wide x (300-600) pixels high.

DONE.

- Synopsis text: Please provide a short standfirst (maximum of 300 characters, including space), limit the bullet points to max. 5 and upload it as a separate .doc file. Please write the bullet points to summarise the key NEW findings. They should be designed to be complementary to the abstract - i.e. not repeat the same text. We encourage inclusion of key acronyms and quantitative information (maximum of 30 words / bullet point). Please use the passive voice.

DONE

18) Source Data: Please ensure that a completed Source Data checklist is uploaded as a Related Manuscript File (you will be contacted by my colleague Hannah Sonntag with further instructions on Source Data and the checklist that needs to be filled out). Source Data should be organized as a single source data file (zipped) per figure for main figures (all EV and/or Appendix figure Source Data can be included in a single folder), with the panels clearly visible in the folder structure instead of a single

excel file for all Source Data. e.g. all the Source data files for figure 1 need to be saved in a single folder and this needs to be zipped and then uploaded as "SD figure 1.zip" file.

DONE, in FAIRDOMHub.org, please see email from Data Editor.

19) As part of the EMBO Publications transparent editorial process initiative (see our policy here: https://www.embopress.org/transparent-process#Review_Process), Molecular Systems Biology will publish online a Peer Review File (PRF) to accompany accepted manuscripts. This file will be published in conjunction with your paper and will include the anonymous referee reports, your point-by-point response and all pertinent correspondence relating to the manuscript. Let us know whether you agree with the publication of the PRF and as here, if you want to remove or not any figures from it prior to publication. Please note that the Authors checklist will be published at the end of the PRF.

Noted. We hope the PRF will include the Review Commons reviews?

20) Please provide a point-by-point letter INCLUDING my comments and your detailed responses (as Word file).

DONE - This file.

Kind regards,

UUG and AJM, on behalf of all authors.

Reviewer #1:

The authors have done a good job in addressing most of my comments. I support the publication of the revised manuscript in MSB.

Reviewer #2:

The authors have addressed my comments appropriately.

Reviewer #3:

The authors have made commendable efforts to address the comments and suggestions provided in the initial review. They have adequately addressed the major concerns, particularly by thoughtfully discussing the possibility that discrepancies between the estimated and actual DNA-binding dissociation constant values may stem from missing regulatory mechanisms in the model. Additionally, the order of the figures has been corrected, and typographical issue has been addressed. The manuscript is now well-organized and I am pleased to recommend its publication in Molecular Systems Biology.

Rev_Com_number: RC-2024-02679

New_manu_number: MSB-2024-12754-T

Corr_author: Millar

Title: Abundant clock proteins point to missing molecular regulation in the plant circadian clock

3rd Jan 2025

Manuscript number: MSB-2024-12754R

Title: Abundant clock proteins point to missing molecular regulation in the plant circadian clock

Dear Dr. Millar,

Congratulations on an excellent manuscript, I am pleased to inform you that your manuscript has been accepted for publication in Molecular Systems Biology. Thank you for your comprehensive response to referee concerns. It has been a pleasure to work with you to get this to the acceptance stage.

Yours sincerely,

Poonam Bheda, PhD
Scientific Editor
Molecular Systems Biology
